# Avalanches and micrometeorology driving mass and energy balance of the lowest perennial ice field of the Alps: a case study

Rebecca Mott[1,2], Andreas Wolf[3], Maximilian Kehl[1], Harald Kunstmann[1,4], Michael
Warscher[1,4,5], Thomas Grünewald[2]

[1] Institute of Meteorology and Climate Research, Atmospheric Environmental Research (KIT/IMK-IFU), KIT-Campus Alpin, Garmisch-Partenkirchen, Germany
[2] WSL Institute for Snow and Avalanche Research SLF, Davos, Switzerland
[3] Institute for Karst and Cave Science, Germany
[4] Institute of Geography, University of Augsburg
[5] Department of Geography, University of Innsbruck, Austria

*Correspondence to*: Rebecca Mott (mott@slf.ch)

**Abstract.**

The mass balance of very small glaciers is often governed either by anomalous snow accumulation, winter precipitation being multiplied by snow redistribution processes (gravitationally or wind-driven), or by suppressed snow ablation driven by micrometeorological effects lowering net radiation and/or turbulent heat exchange. In this case study, we analysed the relative contribution of snow accumulation and ablation processes governing the long- and short-term mass balance of the lowest perennial ice field of the Alps, the Ice Chapel, located at 870 m ASL in the *Berchtesgaden National Park* (Germany). This study emphasizes the importance of the local topographic setting for the survival of a perennial ice field located far below the climatic snow line. Although long-term mass balance measurements of the ice field surface showed a dramatic mass loss between 1973 and 2014, the ice field mass balance was rather stable between 2014 and 2017 and even showed a strong mass gain in 2017/2018 with an increase in surface height by 50 -100% relative to the ice field thickness. Measurements suggest that the winter mass balance clearly dominated the annual mass balance. At the Ice Chapel surface, 92% of snow accumulation was gained by snow avalanching, thus clearly governing the 2017/2018 winter mass balance of the ice field with mean snow depths of 32 m at the end of the accumulation period. Avalanche deposition was amplified by preferential deposition of snowfall in the wind-sheltered rock face surrounding the ice field.

Detailed micrometeorological measurements combined with a numerical analysis of the small-scale near-surface atmospheric flow field identified the micrometeorological processes driving the energy balance of the ice field. Measurements revealed a katabatic flow system draining down the ice field throughout the day, showing strong temporal and spatial dynamics. The spatial origin of the thermal flow system was shown to be of particular importance for the ice field surface energy balance. Deep katabatic flows, that developed at higher-elevated shaded areas of the rock face and drained down the ice field appeared to enhance sensible heat exchange towards the ice field surface by enhancing turbulence close to the ice surface. Contrary, the shallow katabatic flow developing at the ice field surface appeared to laterally decouple the local near-surface atmosphere from the warmer adjacent air supressing heat exchange. Results thus suggest that shallow katabatic flows driven by the cooling effect of the ice field surface are especially efficient in lowering the climatic sensitivity of the ice field to

the surrounding rising air temperatures. Such micrometeorological phenomena must be taken into account when calculating mass and energy balances of very small glaciers or perennial ice fields at elevations far below the climatic snow line.

## 1 Introduction

Very small glaciers are especially sensitive to climatic changes and are considered to be strong climate indicators. Most of the world's smaller glaciers will disappear by 2100 (Radic and Hock, 2011). In the Swiss Alps, 90% of the glaciers have areas of less than 1 km$^2$ (Paul et al., 2001). Similar distributions have been observed for other mountainous regions (Glazirin 1985; Dikich and Hagg, 2004; Bahr and Radic, 2012; Kuhn et al., 2012). Their high number makes them significant contributors to the regional and global hydrological cycle

(Bahr and Radic, 2012) as well as to sea-level rise (Radic and Hock, 2011). Thus, it is crucial to understand the mass balance of very small glaciers and its climatic drivers. Although very small glaciers are especially vulnerable to changing air temperatures due to their small altitudinal extension (Müller, 1988), a large number of very small glaciers and perennial ice fields are known to exist far below the climatic snow line. The reason for their existence was often found to be anomalous accumulation, winter precipitation being multiplied by wind

drift (Dadic et al., 2010), preferential deposition of snow (Mott et al., 2014; Gerber et al., 2017; Gerber et al., 2019; Mott et al., 2018) or avalanches (Kuhn, 1995). Not only snow accumulation in winter, but also regional climate (Mölg et al., 2009; Kaser et al., 2004), convective cloud formation (Nicholson et al., 2013) or micrometeorological effects lowering the incoming solar radiation or changing the turbulent heat exchange (Denby and Greuell, 2000; Escher-Vetter, 2002; Oerlemans and Van Den Broeke, 2002, Petersen et al., 2013)

were found to be driving factors for the survival of very small glaciers and perennial ice fields. The sensitivity of melt rates to temperature change is determined by changes in longwave radiation balance and turbulent heat fluxes (Oerlemans, 2001; Ayala et al., 2015). While the effects of katabatic flows on the mass and energy balance of large glaciers have been intensively investigated (Kuhn, 1995; Oerlemans and Grisogno, 2002, Strasser et al., 2004; Shea and Moore, 2010; Ayala et al., 2015), katabatic flow development over perennial ice

fields and large snow patches (Mott et al., 2015; Mott et al., 2017; Mott et al., 2018) have gained little attention so far. Experimental and numerical studies on the energy balance of perennial ice fields and large snow patches (Marsh and Pomeroy, 1996; Essery et al., 2006; Fujita et al., 2010; Mott et al., 2011; Mott et al., 2013, Curtis et al., 2014, Mott et al., 2016; Schlögl et al., 2018a, b) identified additional micrometeorological processes strongly affecting the local air temperature and associated heat exchange processes, such as cold-air pooling, boundary

layer decoupling and advective heat transport. These processes are also expected to be important climatic drivers for the mass balance of very small glaciers by strongly affecting their sensitivity to an increase in ambient air temperature. Considering the effect of micrometeorological drivers, as well as processes promoting strong amplifications of solid precipitation over very small glaciers (e.g. cirque glaciers) and perennial ice fields will help to improve the assessment of climate change impacts on very small glaciers, representing the majority of

the glaciers in the Alps.

The climatic snow line, which is the line above which snow, will remain all year is about 2500 -2800 m above sea level for the northern slopes of the Alps with a median snow line for the entire Alps of 3083 m +/- 1121 m (Hantel et al., 2012). The lowest perennial ice field of the Alps, the Ice Chapel, is located at approximately 870 m above sea level, which is far below the local and the alpine climatic snow line. The existence of the perennial

ice field is assumed to be attributable to extreme snow accumulation in winter due to avalanches and preferential deposition of precipitation in the wind-sheltered area surrounding the ice field. We assume that the topographic setting of the Ice Chapel further involves micrometeorological effects on snow ablation such as strong topographic shading and the development of thermal flow systems. This study is the first attempt to address snow accumulation and ablation processes affecting the mass and energy balance of the Ice Chapel, located in the *Berchtesgaden National Park*.

We present data on the long-term areal change of the ice field between 1973 and 2018. Mass balance measurements in 2017/2018 allowed us to investigate the winter and summer surface change in very high spatial resolution. Based on those data we estimate the relative contribution of avalanching and precipitation to the winter snow accumulation at the perennial ice field. We further discuss the contribution of micrometeorology on the summer mass balance of the Ice Chapel by experimentally and numerically investigating the flow field development and associated small-scale air and surface temperature variations at the Ice Chapel and its surrounding. The respective results are discussed and summarized at the end of the paper.

## 2 Methods

### 2.1 Study area

The *Berchtesgaden National Park* is located in the Bavarian Alps at the border between Germany and Austria (47.552778°, 12.975833°) and comprises an area of 208 km$^2$. Within this area, the landscape ranges between 501 m and 2713 m a.s.l.. The perennial ice field, the Ice Chapel, is located in the interior of Berchtesgaden National Park (Germany) at the upper end of the Eisbach valley (Figure 1). Located at an elevation between approx. 870 to 1100 m ASL, approximately 1000 m below the regional tree line and well below the actual climatic snow line at the northern slopes of the Alps (approximately 2500 - 2800 m), the Ice Chapel is the lowest perennial ice field of the Alps (Hornauer and Eichner, 1997; Wolf, 2007; Rödder et al., 2010). The perennial ice body is surrounded by the steep rock walls of the Watzmann massif (2713 m). The Watzmann Eastface is on average 51° inclined and some parts of the rock face show slopes above 70° (Figure 1). Due to its steepness and it´s funnel-shaped configuration, the rock face surrounding the Ice Chapel constitutes an avalanche release area of 1,6 km$^2$ with avalanches accumulating in the angle of the rock face, where the Ice Chapel is located. The Ice Chapel is bordered by the rock face to the north-west and two moraines to the north-east and south-west. We refer to moraines as any rocks that have been moved in, on top of, or under the ice field have been deposited close to it. The small Ice Chapel (Figure 1) is located north-east to the Large Ice Chapel and is also fed by avalanches in winter. Contrary to the Large Ice Chapel, the Small Ice Chapel totally disappeared in some years of low snow accumulation.

### 2.2 Remote sensing measurements

### 2.2.1 Mass balance measurements

High-resolution surface measurements were conducted with a terrestrial laser scanner (TLS; Riegl VZ-6000) on 26 October 2017, 17 March 2018 and 29 September 2018 in order to obtain winter and summer mass balance of the Ice Chapel.

The second summer mass balance measurements were done in late September since a snow fall event was forecasted for early October. Since the Ice Chapel receives nearly no shortwave radiation in autumn (Figure 2), no strong ablation was expected for October.

In past studies, repeated TLS was successfully applied to calculate snow volumes (Grünewald et al., 2018) or
snow depth changes during the accumulation (Mott et al., 2010; Schirmer et al., 2011; Sommer et al., 2015) and ablation season (Grünewald et al., 2010; Egli et al., 2011; Mott et al., 2011; Schlögl et al., 2018) with a vertical accuracy of less than 10 cm for 300 m distance (e.g. Prokop et al., 2008; Revuelto et al., 2014). A more general description of the TLS measurement setup and accuracy over snow can be found in Prokop et al. (2008), Schaffhauser et al. (2008), and Grünewald et al. (2010). To reduce scan shadows the laser scanner was set up at
up to three different positions. The area of the Ice Chapel and its surrounding was then recorded with a frequency of 300 kHz and angular step widths between 0.002 and 0.05 depending on maximum measurement distance which ranged from 300 to 500 m. We followed the post processing procedure described by Grünewald et al. (2018 and 2019): First coarse registration was performed using small reflector plates mounted in the area and/or topographic features (such as well-defined rocks) as tiepoints. This registration was then improved by
applying a 3D-surface matching function (Multi station adjustment; Riegl, 2011). In the following, the data were transformed to a global coordinate system (UTM). Finally, data amounts were reduced by aggregation of the point clouds to 25 cm 3D grids (octree filter) and raster of surface change (cell size 0.5 m) were calculated in ArcMap 10.2.

**2.2.2 Estimation of aerial changes since 1973**

The spatial extent of the Ice Chapel was assessed using different remote sensing measurement techniques. The accuracy and point density of each measurement device is summarized in Table 1. The technological progress has allowed for a change from point measurements retrieved from tachymetry (1994) to high resolution 3-D scans of the surface using TLS (since 2007).
Aerial photos (WILD RC7) generated in autumn 1973 allowed for the retrieval of the spatial extent of the Ice Chapel surface with a high accuracy of 0.1 m. Aerial stereo image pairs were assessed with mechanical optical autographs (WILD, A4). In 1994, a geodetic network was created in the western part of the upper Eisbach valley basin in the surroundings of the Ice Chapel, and connected to the national surveying network. In autumn 1994, an engineering theodolite and reflector lot was used to retrieve the surface of the Ice Chapel, complemented with
field sketches. Using tachymetry, the position (x) and altitude (y) of several points of the surface were measured with an accuracy of 0.01 m in both directions. In autumn 2007, measurements of the Ice Chapel surface were conducted combing tachymetry (LEICA, TCRA 1101) and terrestrial laser scanning (LEICA, HDS2500). Further tiepoints were installed at the Ice Chapel surface and surveyed with tachymetry and transformed to the national surveying network. The TLS allowed for the first time to assess the ice field surface with a much a
higher spatial resolution (point density of 0.1 m for 100 m measurement distance) and with a higher measurement accuracy of 0.005 m. Following the same procedure as in 2007, the Ice Chapel surface was surveyed in autumn 2014 using a FARO scanner (FOCUS3 DX130) and a LEICA theodolite (TCRA 1200) and in autumn 2017 using a total station TRIMBLE (SX10) and a TLS (Riegl VZ-6000). In autumn 2018, surface measurements were conducted with the Riegl VZ-6000 as described above.


Table 1: Overview on remote sensing measurements applied at the Ice Chapel since 1973

| Time of measurement | Measurement technology | Device | Accuracy x, y (m) | Point density (cm) | Measured surface area [$m^2$] |
|---|---|---|---|---|---|
| September 1973 | Photogrammetric analysis of aerial photos | WILD RC7 | 0.1* | | 44 100 |
| October 1994 | Tachymetry | Theodolite WILD, TC 1610 | 0.01* | | 34 300 |
| July 2007 | Tachymetry | Theodolite LEICA, TCRA 1101 | 0.01* | | 40 000 |
| | Terrestrial laser scanning | LEICA, HDS2500 | 0.005* | 10 ** | |
| November 2014 | Terrestrial laser scanning | FARO, FOCUS3 DX130 | 0.005* | 6.14 ** | 12 500 |
| | Tachymetry | Theodolite LEICA, TCRA 1200 | 0.01* | | |
| October 2017 | Terrestrial laser scanning | Total station TRIMBLE, SX10 | 0.005* | 12.28 ** | 13 000 |
| | Terrestrial laser scanning | Riegl VZ 6000 | 0.015* | 2 ** | |
| | Tachymetry | Total station Trimble, SX10 | 0.01* | | |
| September 2018 | Terrestrial laser scanning | Riegl VZ 6000 | 0.015* | 2 ** | |

* according to manufacturer information for specific measurement setups

** values are given for 100 m measurement distance

**2.3 Meteorological measurements: air temperature and surface temperature measurements**

The spatial variability of near-surface air temperatures in the region of the Ice Chapel was captured measuring air temperatures at 20 locations using 4 mobile meteorological stations SnoMoS (Pohl et al., 2014). We simultaneously measured air temperature and humidity at 2 m above the surface with mobile towers on October 26, 2017 between 12:00 PM and 01:00 PM, along three transect lines: (1) the lower part of the Ice Chapel and adjacent slope of the north-easterly moraine, (2) the downstream region of the Ice Chapel snout and (3) on top of the south-westerly moraine. Directly over the Ice Chapel, only two measurements were possible due to safety reasons. Measured air temperatures are 10 minute averages. One station was installed over the entire measurement period of one hour at one location further downstream to take into account the temporal evolution of air temperature. Air temperature at this location was rather constant during the measurement period, varying by less than 1 °C.

A thermal infrared camera (IR camera hereafter), VarioCAM HD research 900 (Infra Tec GMBH), was used to measure surface temperatures at the Ice Chapel area with a high spatial and temporal resolution. The camera uses an uncooled microbolometer array for the detection of thermal infrared radiation in the spectral range of 7.5–14 μm. The resolution of the camera is 1024 × 768 pixels with a measurement range from −40° to 1200°C and an accuracy of ±1.5 K for the measurement range. Measurements were conducted during 2 days with partly cloudy conditions (12 and 13 July 2018) and 2 days with clear sky conditions (19 and 20 July 2018). High resolution

data were acquired on an hourly basis from 09:00 AM until 05:30 PM. The camera position slightly moved over the course of three measurement days due to instabilities of the tripod, making an analysis of the temporal evolution of single pixels unfeasible. Since the presence of clouds significantly influenced the spatial variability of surface temperatures during 13 July, we focus the analysis on days with clear sky conditions.

No meteorological data are available at the Ice Chapel area during the IR measurements. We only analyse the spatial variability of surface temperatures (TS) over the same surface type (rock and debris; excluding ice surface) and the change of patterns over time. We are thus only interested in relative temperatures and not in absolute surface temperatures. We also limit the analysis to single profile lines of rather small distances of tens of meters in order to minimize differences in measurement errors due to not applying atmospheric corrections. In

an earlier study (Mott et al., 2017), a comparison of relative temperatures obtained from the IR camera used here, a CNR4 sensor and a mobile weather station showed that relative values coincide well, mainly not exceeding the arbitrary threshold of ±0.5 K. Since our analysis is limited to relative changes of surface temperatures in space and time, we do not apply corrections for the emissivity of the different surfaces and the atmospheric transmissivity which mainly affect absolute values or relative values for very large areas with

strongly varying distances. The dynamics of air flow is derived from the change in surface temperatures along transect lines crossing the north-easterly moraine adjacent to the lower part of the Ice Chapel and crossing the south-westerly moraine at the outflow region of the Ice Chapel snout (see Figure 9). Each transect line represents the average values of five neighbouring pixels.

Table 2: Measurement campaigns conducted in 2017 and 2018

| Measurement campaigns 2017/2018 | Measurements |
|---|---|
| 26 October 2017 | TLS, mobile meteorological stations |
| 17 March 2018 | TLS |
| 12 , 13, 19, 20 July 2018 | IR Measurements |
| 29 September 2018 | TLS |

**2.4 Atmospheric modelling of near-surface boundary layer development**

An atmospheric model Advanced Regional Prediction System (ARPS) was applied to simulate the atmospheric flow field in the region of the Watzmann massif, including the area of the Ice Chapel. Flow fields were

calculated on a horizontal resolution of 20 m, on a domain covering the entire Watzmann massif (Figure 1). The simulations use 60 vertical terrain-following levels. The vertical resolution of the first grid above surface ranges between 1.4 m and 2.6 m with an average value of approximately 2 m at the ice Chapel area.

We used a small integration time step of 0.01s and an acoustic wave mode with a time step of 0.001s. Flow fields were calculated for 19 July (measurement campaign in 2018) and 26 October (measurement campaign in

2017) reflecting two different situations regarding the exposure of the rock face and the Ice Chapel to shortwave radiation. While most of the simulation area is sun-exposed during the day in July, it gets sun-shaded during most of the day in October (Figure 2). Simulations start at 12 UTC and were run for an integration time of 3600 s. We chose noon, because approximately peak radiative forcing should highlight the difference in boundary layer development between the two different situations in the ablation season. The results on boundary layer

development and heat exchange processes are a snapshot in time and do not cover the temporal variability in turbulent fluxes of heat and momentum that is connected to large eddies. Running the model over an entire day generates numerical stability problems which are likely related to insufficient vertical resolution when shallow stable atmospheric layers develop (Mott et al., 2015). Results of Raderschall et al. (2008) could evidence fully

developed turbulent flow field characteristics after an integration time of about 600 s and 3600 s when running idealized simulations with ARPS using a similar domain size, model resolution and a small integration time step of 0.01 s.

Air temperature and wind velocity measurements are only available in a high spatial resolution in the area of the Ice Chapel on 26 October. For model initialization, meteorological data from three stations have been

considered: permanent weather station Watzmann Grat (1635 m), permanent weather station Kühroint (1420 m) and mobile meteorological stations at the moraine close to Ice Chapel (900 m). Air temperatures obtained from the stations Kühroint, Watzmann Grat and the mobile station at the moraine Ice suggest two different model setups to analyse the flow field development for July and October: (1) slightly stable atmospheric conditions for October simulations (Väisälä frequency N ≈ 0.01 s−1) and (2) neutral atmospheric conditions for July

simulations. Using measured air temperatures from stations at different elevations  The first setup is considered to reflect the situation of strong sun-shading of the lower part of the east face of the Watzmann massif and no solar radiation in the area of the Ice Chapel over the entire day favouring stable atmospheric conditions. The second setup reflects a situation of a mostly sun-exposed Watzmann Eastface and sun-exposed Ice Chapel, which is typically found earlier in the summer season (e.g. 19 July, Figure 2). We used this methodology since

no direct measurements are available at the Watzmann Eastface and no meteorological measurements are available at the Ice Chapel for 19 July. Furthermore, heating of the sensors by shortwave radiation might also affect air temperature measurements. Initial atmospheric stability is thus only an approximation of local atmospheric conditions. Since simulations are not run for 24 hour integration time the integration time does not allow for the full adaptation of the near-surface air field to the daily cycle of radiation. As discussed above,

however, we expect the flow field to adapt to thermal forcing during the integration time, also changing the local atmospheric stability, in particular over the ice field surface.

We used the same initial wind velocities, air temperatures and surface temperatures for both set-ups as no meteorological measurements are available at the Ice Chapel area on 19 July and for comparability of numerical results on flow development driven only by differences in radiation. Initial air temperature and wind velocity

were obtained from mobile measurements at moraine Ice Chapel for 26 October. The initial surface temperature of snow-free areas was set to 11.6°C, as measured at station Kühroint (Figure 2). The initial surface temperature of the Ice Chapel surface was set to 0°C.

## 3 Results and Discussion

### 3.1 Long-term surface area change between 1973 - 2017

Continuous measurements of the ice-field surface between 1973 and 2018 have been performed using different remote sensing methods (see Section 2.2.2). Measurements of the ice surface in the autumn of 6 individual years (Table 1) show that the minimum surface area at the end of the ablation period decreased from 44100 $m^2$ in 1973 to 12500 $m^2$ in 2014 with a slightly increasing area between 2014 and 2017 (+ 677 $m^2$) and a significant increase between 2017 and 2018. The larger extent measured in 2007 is due to the earlier date of measurement (July). At

this early time in the ablation season the surface of the Ice Chapel was even smaller than measured in late

September 1973, more than two months later in the ablation season, emphasizing the tremendous retreat of the ice field during the last thirty years.

While the planimetric area of the ice field was aligned in E-W direction in 1973 (main axis of slope direction), with a length of almost 500 m, the extent in N-S direction was less than 400 m. Within the last thirty years the length in E-W direction was strongly reduced to approximately 200 m. Strongest reduction of the surface area is shown at the upper and lower boundaries (lower and higher elevations) where the thickness of the ice field was smallest (Figure 4). The reduction of the surface area in N-S directions was much smaller, still featuring a length of around 230 m. This is most probably due to the larger thickness of the ice body there. The profiles shown in Figure 4, indicate a rather small change in surface height between 1973 and 1994 indicating a small total mass loss of the Ice Chapel, but a rapid decrease between 1994 and 2014 with more than 30 m change in surface height at the main body. Although the surface area increased between 2014 and 2017, measurements evidence a decrease in surface height (few meters) and a total volume loss of 56500 $m^3$ during these three mass balance years. This shows that the change in surface area between 1994 and 2017 was connected to a massive volume loss during the last two decades.

**3.2 Measurements of snow accumulation, ablation and total change of the ice field surface in 2017/2018**

**3.2.1 Snow accumulation October 2017 – March 2018**

The map of snow accumulation in winter 2017/2018 was obtained from terrestrial laser scanning before winter on 16 October 2017 and at the end of the winter on 17 March 2018 (Figure 6). Maximum seasonal snow depth at the station Kühroint was measured on 8 March 2018. At the end of the accumulation season, TLS measurements evidenced between 25 and 40 m of snow accumulation at the main body of the Ice Chapel. Largest snow accumulation was measured at the upper boundary of the ice body close to the rock wall, with locally more than 40 m of snow deposition. The lower elevated area of the Ice Chapel gained between 20 and 39 m of snow, with decreasing snow accumulation towards the lateral boundary of the Ice Chapel. The mean snow accumulation rate measured at the main body of the Ice Chapel is 32.4 m (Figure 6, grey rectangle), which is 3000% of the snow depth measured at the flat-field site Kühroint (Figure 1), showing a snow depth of 1.07 m at this day and a maximum snow depth of 1.39 m at time of peak accumulation (9 days earlier) (Figure 5). The large snow depositions at the Ice Chapel could be clearly identified as avalanche deposits (Figure 1d).

Clearly above average snow deposition due to avalanching was observed up to a distance of 200 m downstream of the maximum extent of the Ice Chapel in autumn 2017. The Small Ice Chapel showed similar snow accumulation rate with approx. 20 to 35 m of snow deposition. With less than 0.5 m snow accumulation, most areas of the lower part of the rock face above the Ice Chapel show much smaller amounts of snow deposition. Significant snow deposition can be found at the rock face in gullies and avalanche pathways with snow heights between 5 to 10 m, which can be mainly attributed to snow avalanches and snow slides. Areas, which are not prone to avalanches, such as the moraine shoulders and slopes in the surrounding of the Ice Chapel, can be clearly distinguished in the HS maps from areas where snow deposition is clearly dominated by avalanches. The comparison between mean values in areas not affected by avalanches and areas clearly affected by avalanche deposition, show that approximately 92 % of snow accumulation at the main body of the Ice Chapel can be attributed to avalanching from the large avalanche catchment of the Watzmann Eastface. Only 8 % of the winter snow accumulation at the Ice Chapel can be ascribed to solid precipitation. However, part of snow deposition in the surrounding area of the ice field could also be influenced by depositions from powder snow avalanches. The

estimated values are similar to values published in an earlier study of Rödder et al (2010) who found a 90 % contribution of avalanches to snow deposition at the Ice Chapel for winter 2006/2007.

Areas where snow accumulation is solely attributed to solid precipitation show snow depths between 1 and 4 m with a mean snow depth of 2.5 m (Figure 6, red rectangle) which is 250% of the snow depth measured at the flat

field site Kühroint (Figure 1). Due to its funnel-shaped configuration, the Watzmann Eastface is sheltered from strong winds for most prevailing wind directions (Warscher et al., 2013). The associated reduction of wind speed downstream of the ridge crests and the associated stream-wise flow convergence over the large leeward slopes are assumed to promote higher precipitation rates over the entire wind-sheltered cirque due to preferential deposition of precipitation (Mott et al., 2014). Finally, the high avalanche activity in the area leads to a strong

amplification of snow deposition at the Ice Chapel, especially in winter with high snowfall rates. It can thus be assumed that the relative contribution of avalanching to total winter snow deposition at the Ice Chapel increases with increasing winter precipitation. Due to wind-sheltering and associated low wind velocities in the entire investigation area (Warscher et al., 2013), we assume that snow drift is of minor importance for the winter mass balance of the Ice Chapel.

### 3.2.2 Snow ablation March 2018 – September 2018 and  total surface change during the mass balance year 2017/2018

|  | Range | Mean |
|---|---|---|
| Snow accumulation | 25 - 40 m | 32.4 m |
| Snow ablation | 15- 19 m | -17.5 m |
| Net surface change October 2017 – September 2018 | (+) 6 - 20 m | +15.7 m |

Maps of snow ablation during summer 2018 are obtained from terrestrial laser scanning at the start (17 March 2018) and the end of the ablation season (29 September 2018) (Figure 7 a). The difference between the measured surface in October 2017 and September 2018 provided the total surface change during the mass balance year 2017/2018 (Figure 7 b).

Measurements evidenced a positive mass balance of the Ice Chapel at the end of the ablation season (September

2018) with a total increase in surface height ranging from 6 m (at the ice field snout) to 20 m (central part of the ice field). The thickness of the ice body increased by approximately 100 % at lowest elevations and by approximately 50 - 70 % at the central part (Figure 6 b, Figure 4). Also the length of the ice field increased by approximately 70 m, which is visible by the downward shift of the ice field snout (Figure 7).

Since the entire ice field surface was still covered by seasonal snow in late September, the change in surface

height at the end of the ablation season is mainly limited to snow ablation. No ice ablation could be evidenced. Snow ablation rates ranged between 15 and 19 m with highest ablation at the lower-elevated and the high-elevated parts of the Ice Chapel. A distinct area of lower ablation in summer was measured at the central part of the ice field, where maximum increase in net surface height was observed. Lowest net surface increase was found at the lower-elevated areas of the Ice Chapel, because of the strong snow ablation combined with less

avalanche deposition in winter (Figure 7 b).

Contrary to mass balance measurements of large glaciers in the Alps, revealing negative mass balances since 1970 (Kaser et al., 2006), the survey of the perennial ice field did not reveal a continuing decrease from year to year, but showed some years with an increasing net surface area (Figure 3), such as in 2017 and 2018 and even a clearly positive mass balance in 2018 (Figure 7). These measurements highlight the ability of the Ice Chapel to

feature significant mass gain during individual years with strong precipitation in winter driving avalanche deposition at the Ice Chapel. Such an increase in surface height by more than 50% in one mass balance year is only possible for topographic locations featuring anomalous snow accumulation. The positive mass balance is connected to above average precipitation in winter (Figure 5) and related avalanche deposition at the Ice Chapel and to a long snow cover duration, as presented by Rödder et al. (2010). While winter snow accumulation was

small between 2013/2014 and 2016/2017 (Figure 5), winter snow accumulation was above-average in winter 2017/2018, explaining the positive mass balance for this year. A late melt out of the snow cover in the ablation season favours a positive mass balance of the ice field by lowering the surface albedo and thus lowering net shortwave radiation entering the Ice Chapel.

**3.3 Measured and modelled micrometeorology at the Ice Chapel: katabatic flow and associated effects on near-surface air temperatures and turbulent heat exchange**

### 3.3.1 Measured air temperature field

Maps of air temperatures (TA), measured at 2 m above ground, are presented in Figure 8 for 26 October 2017,

12:00 AM -01:00 PM.  At this time, the entire area of the Ice Chapel and its surrounding was shaded from sun and no clouds were present. Under uniform solar radiation, measurements evidence a very high spatial variability of TA at the Ice Chapel and downstream of the ice field. Measurements showed values of about 8.5 °C at the lowest part of the Ice Chapel and a TA minimum of 6.9 °C measured a few meters downstream of the ice field snout where cold air flow exits the ice-field body. Downstream of the ice field, air temperatures were lowest at

the bottom of the moraine slopes and continuously increased along the slopes with maximum TA of 12.7 °C at the top of the moraine, resulting in an air temperature gradient of 1.5 °C at 10 m distance. Measurements of relative humidity (not shown), also reveal much higher relative humidity with values ranging between 80 to 90% close to the ice field snout and 76% above the ice field surface. Relative humidity was much lower at the top of the moraine with values of about 60% indicating much drier air there. Measurements thus suggest the presence

of two different cold air flows at the Ice Chapel with different origins. A cold and humid air flow originating from the caving system of the Ice Chapel and a katabatic flow evolving at the Ice Chapel surface, both draining down the gully downstream of the ice field. Contrary, the upper part of the moraine appeared to be not affected by a katabatic flow system at this time of the day.

**3.3.2 Measured surface temperature fields**

Surface temperature measurements have been conducted using an IR camera during four days in July 2018 in order to obtain an indirect measure of small-scale micrometeorology in the area of the Ice Chapel. Resulting surface-temperature (TS) maps are presented in Figure 9 for 19 July 2018. Absolute values of TS might include uncertainties due to corrections not applied to the data. We thus only discuss the relative values of surface

temperatures in space and time as earlier studies showed a high accuracy of relative values of IR measurements (Grudzielaneck et al., 2015; Mott et al., 2017). As the Ice Chapel surface was at its melting point throughout the

measurement days, the analysis of changes in surface temperatures was limited to the ice-free surrounding area, mainly featuring debris and rock. Transect lines (L1, L2) are shown for different points in time during two days (12 and 19 July) revealing the spatial and temporal dynamics of TS along the moraine slopes (Figure 10 a-d). Note that the camera position slightly moved over the day on 12 July 2018 due to instabilities of the tripod. The standard deviation of TS during the day, reflecting the cooling and warming rates of surface pixels over the day, are calculated from measurements conducted at 6 different measurement times (Figure 9 a -d), is given in Figure 10 (e, f) for 19 July. This was the only day without shifts of the images during the day and without influences of local cloud formations on local TS. Potential incoming radiation for the respective area is shown for 19 July in Figure 2 b and Figure 11. Contrary to the measurement day in October, most of the area was exposed to the sun from 10 AM until 2 PM with small spatial variations in the potential incoming solar radiation at the Ice Chapel and its surroundings (Figure 11). In the morning hours parts of the rock face were shaded receiving no direct shortwave radiation. After 2 PM, the south-westerly moraine received less radiation than the easterly one. However, radiation is spatially consistent along the individual moraine slopes during the entire measurement days. We thus expect that small-scale spatial differences in surface temperatures along the moraine slopes can be mainly attributed to differences in the turbulent heat exchange between the surface and the atmosphere, mainly driven by small-scale dynamics of the atmospheric boundary layer flow.

Maps presented in Figure 9 highlight the high spatial variability of TS during the course of the day in the surrounding area of the main body of the Ice Chapel. Air temperatures measured at Kühroint (1420 m) ranged between 17 and 18 °C. Maxima of TS are revealed at the flat areas at the top of the moraines. TS are declining with decreasing distance to the ice surface (e.g. area at L1). Downstream of the Ice Chapel (area at L2, Figure 9) TS maps obtained in July show very similar spatial patterns as observed and modelled for the 2 m TA in October (Figure 10): These patterns are characterized by strong temperature gradients of several degrees along the moraine slopes (Figure 9). Surface temperatures persist at significantly lower values in the entire downstream area at the bottom area of the gully. Transect lines L1 shown for certain points in time (Figure 10 a, c) indicate a very strong suppression of TS in the lowest few meters of the moraine slopes above the Ice Chapel. In this area, measurements evidence a strong temperature gradient of up to 2.5 °C change in surface temperature per pixel (approx. 0.2 -0.3 m resolution) with strongly increasing surface temperatures with increasing distance to the ice surface. A local increase of surface temperatures of several degrees at the surface of the Ice-Chapel mark an area of debris cover at the snow surface (Figure 10 c). As discussed above, radiation effects cannot explain the strong spatial differences in daytime surface temperatures along the slopes. We thus assume that a thermal flow draining down the ice field and downstream gullies is cooling the surface at the lower parts of the moraine slopes during the day. As discussed above, we could also observe such a drainage flow during high-resolution air temperature measurements in October (Figure 10). The existence of the ice field throughout the year and partial shading of the very steep rock face suggest the presence of the katabatic flow system over the entire ablation season.

Transect line L2 (Figure 10 b, d) reveals the influence of two flow regimes already discussed for the air temperature fields. At the gully downstream of the snout of the Ice Chapel, surface temperatures are affected by the cold air outflow from the Ice Chapel caving system. Surface temperatures are up to 10°C colder than measured a few meters above. With increasing height above the gully surface, surface temperatures show large spatial gradients due to the influence of the drainage flow originating from the Ice Chapel surface with significantly warmer air temperatures than the cold air outflow below. The drainage flow at the ice field surface

appeared to be well-developed in the morning (10 AM, Figure 10b and Figure 11), originating from the shaded parts of the rock face draining down the shaded areas of the large and Small Ice Chapel (Figure 9 b, Figure 11 a) and merging at the north-easterly moraine. The well-developed katabatic wind appeared to persistently cool the surface in the morning hours by advecting colder air from the rock face, which is also visible in Figure 10 (b, d)

with much colder morning surface temperatures at the north-easterly moraine than on the south-westerly moraine. At noon, surface temperatures are balanced between both moraine shoulders, most probably due to an attenuation of the drainage system, limiting the drainage flow to the lowest few meters above the gully surface. These strong differences in surface temperatures are clearly visible for both days presented in Figure 9 and 10. Standard deviations of TS are strongest at the upper parts of the moraine slopes reflecting a significantly stronger

increase of surface temperatures until late afternoon than at the lowest few meters above the Ice Chapel. Surface temperatures increase between 10 AM and 3 PM by about 10 – 17°C at the upper sections of the moraine slopes, peaking at the moraine shoulders. At the lower parts of the moraine slopes, surface temperatures increased by only 3 to 6°C indicating a suppression of daytime surface heating in areas affected by the drainage flow (Figure 10) resulting in a significantly lower standard deviation of TS (Figure 10 e, f). Standard deviation of TS is

especially small at the debris covered location at the Ice Chapel surface (Figure 10 c, e) indicating a strong cooling effect of the drainage flow close to the ice surface. Not only surface heating is suppressed by the presence of the drainage flow, but also the cooling of the surface after sunset. As soon as the area gets shaded, surface temperatures drop by 9°C (L1) and 7 - 9°C (L2) at areas assumed to be not affected by drainage flows but only by 3 - 6°C (L1, L2) at the lower parts of the slopes. We explain the extenuated surface cooling by

higher turbulent heat exchange in these areas forced by higher near-surface wind velocities. Similar effects of the katabatic flow are expected to take place at the ice field surface, attenuating the warming of near-surface air temperatures during the day but also attenuating the cooling of ice surface during night.

### 3.3.3 Modelled air temperature and wind field

Simulation results obtained from the atmospheric model ARPS using two different model set-ups (see Section 2.4) show near-surface air temperature fields characterized by large horizontal temperature gradients over the ice field area with coolest air temperatures at the lowest downwind part of the ice field and at the bottom of the moraine slopes (Figure 12 a, c). Spatial patterns of air temperatures are similar to measured patterns of air temperature (Figure 12) and surface temperature (Figure 9, 10) but show smaller spatial air temperature

gradients. The development of the air temperature field differs between the two model setups, mainly driven by differences in flow field development. For both model setups, a katabatic flow is present over the Ice Chapel but with differences in the development such as the onset, the depth and the magnitude of wind velocity maxima of the drainage flow (Figure 13).

Model results show that in case of a sun shaded rock face ((Figure 12 a, b, set-up 1), the cold air drains down the

very steep slopes resulting in a well-developed katabatic flow draining down the Ice Chapel with a wind speed maximum ranging between 7 – 9 m above the ice surface (Figure 13). Air temperatures fields during such a situation were measured at 12:00 PM, 26 October (Figure 8, Figure 12). Signatures on the surface temperatures of such deep and well developed katabatic flows have been observed by IR images during the morning and late afternoon when the lower parts of the rock face are shaded from sun. In case of well-developed katabatic flows,

the flow speeds up over the cold surface of the ice field further decreasing near-surface air temperatures over the lower part of the Ice Chapel and downwind of the perennial Ice field. In the area of the ice field, air temperatures

range from 7.8 °C to 12.4 °C. The cold air flow exiting the Ice Chapel caving system at the snout, which was observed by measurements, is not captured by the model. This is one reason why the modelled temperature range downstream of the Ice Chapel is smaller than the measured one with minimum values higher than evidenced by measurements. Also, the horizontal distance between minimum and maximum air temperature increases from 40 m evidenced by measurements to 160 m in the model, most probably caused by the overestimation of the depth of the katabatic flow by the model due to resolution restrictions. The representation of the katabatic flow depth is known to be strongly dependent on the near-surface vertical grid resolution (Mott et al., 2015). A vertical resolution of approximately 2 m close to the surface appear to be too coarse to capture shallow katabatic flows which typically have a jet maximum of less than 2 m above ground (Denby, 1999; Oerlemans and Grisogono, 2002).

Model results of setup 2 (Figure 12 c, d) suggest that in the absence of katabatic flow development at the rock face, the cooling of the near-surface air by the ice surface results in the onset of a very shallow katabatic flow directly over the perennial ice field with an increase in wind velocity with increasing fetch distance over snow/ice (Figure 13). Similar to setup 1, maximum wind velocities are found downwind of the Ice Chapel, coinciding with minimum air temperatures. At the central part of the Ice Chapel, near-surface wind velocities are, however, significantly smaller compared to the well-developed katabatic flow revealed by setup 1. Due to the much shorter fetch distance of the katabatic flow, the depth of the katabatic flow (Figure 13) and also the horizontal stretching of the flow is much smaller than for setup 1. Similar to measurements, air temperatures reach a minimum at the lowest part of the Ice Chapel and at a distance of 150 m downstream of the ice field snout.

### 3.3.4. Indications of flow field development on the heat exchange and snow ablation patterns

The onset and development of the katabatic flow appears to have a significant influence on the turbulent heat exchange at the ice surface (Figure 14). For situations with a well-developed katabatic flow (set-up 1), near-surface wind velocities of more than 4 m/s and positive near-surface temperature gradients result in a strong turbulent mixing and strong downward sensible heat flux with maximum values along the entire Centre line of the ice field. Contrary, in case of a shallow katabatic flow that developed directly over the ice field (situation 2), wind velocities are much lower at the upper and central parts of the ice field coinciding with a smaller turbulent sensible heat exchange there. Increasing wind velocities in downwind distance involved increasing downward sensible turbulent heat exchange towards the lower parts of the ice field because of the approximately linear dependence of turbulent heat exchange on the wind velocity (Dadic et al., 2013). Although turbulent fluxes are locally higher at the lower parts due to higher mean near-surface air temperatures than for the well-developed katabatic flow situation, the average turbulent sensible heat flux at the ice field surface is calculated to be significantly smaller (Figure 14). This confirms earlier results presented by Mott et al. (2015) showing stronger turbulent heat fluxes in situations with well-developed katabatic flows compared to shallow katabatic flows. The strong katabatic winds enhance mechanical turbulence close to the surface and remove the shallow stable layer close to the ice surface that typically promote a suppression of turbulent heat exchange. Modelled mean turbulent heat fluxes at the ice field surface (Figure 14) were smaller in case of weaker and more shallow drainage flows due to a decoupling of the atmospheric layer adjacent to the ice field surface from the warmer air above. These model results are similar to results discussed in Mott et al. (2015) who emphasized the isolation effect of shallow

katabatic winds over large snow fields, also referred to as lateral atmospheric decoupling. Model results suggest that in such situations, the snow and ice melt is only marginally affected by higher ambient air temperatures. Since the analysis on the temporal and spatial evolution of surface temperatures indicated a dominance of the shallow katabatic flow in summer, we expect the isolation effect of the katabatic flow to strongly affect the mass balance of the ice field during the ablation season. Snow ablation measurements (Figure 7 a, b) evidenced smallest ablation rates at the central part of the ice field and a strong increase in ablation rates with downwind distance. The lateral decoupling effect of the shallow katabatic flow might partly explain these ablation patterns with lower sensible heat fluxes at the central part of the ice field and the increasing heat exchange in downwind distance. The effect of lateral decoupling also coincides with the observed attenuating effect of the katabatic flow on the heating and cooling rates of surface temperatures. Maximum snow ablation rates at high elevated areas, however, cannot be explained by modelled flow field dynamics. One reason for above-average snow ablation in this region might be the larger amount of debris accumulated at the boundary areas adjacent to the rock face and the moraine slopes. Other effects could be strong longwave radiation from the surrounding rock face and stronger subsidence of the surface at the upper boundaries of the ice field where the lateral crevasses are most pronounced.

## 4. Conclusions

This study presents a detailed investigation of the relative contribution of accumulation and ablation processes driving the mass balance of the lowest perennial ice field of the Alps, i.e. the Ice Chapel, located far below the climatic snow line, making it to a popular tourist attraction. The long-term monitoring of the surface change of the ice field during the last three decades, applying different remote sensing techniques, evidenced a dramatic decrease of the ice field area between 1973 and 2014 but a steady mass balance between 2014 and 2017 and a strong increase in surface height between 2017 and 2018. In order to analyse different factors allowing the survival of the perennial ice field till today, we presented a detailed analysis of the winter and summer surface change of the ice field in 2017/2018. High resolution TLS evidenced an increase in surface height of the ice field by 14.5 m in average (around 50% - 100% increase relative to ice body thickness) and an increase in length by around 25 %. The experimental and numerical analysis suggested a clear dominance of snow avalanches on the winter mass balance, contributing 92 % to the total snow deposition at the Ice Chapel with a mean snow depth of 32.4 m. Only 8% of the total winter snow accumulation was gained by snowfall at the ice field. These results clearly suggest that the orographic setting allowed the perennial ice field to survive at such a low elevation, by gaining 3000% of winter snow accumulation compared to a near-by flat field snow station. These findings emphasize the tremendous effect of avalanching on the survival of very small glaciers or perennial ice fields which are often located far below the climatic snow line. Precipitation via preferential deposition in this wind-sheltered area only indirectly contributed to the mass balance of the ice field by resulting in above-average snow accumulation in the avalanche release area of 1,6 km$^2$ of the 2000 m high Watzmann Eastface, amplifying avalanche deposition at the ice field. This study thus suggests that the existence of the perennial ice fields is due to anomalous winter accumulation at those locations, winter precipitation being multiplied by preferential deposition of snowfall and/or redistribution of snow by avalanches.

Earlier studies on the mass balance of very small glaciers at low elevations also suggested that the respective orographic setting typically promote micrometeorological processes suppressing ablation in summer (e.g. Denby

and Greuell, 2002; Oerlemans and Van Den Broeke, 2002, Petersen et al., 2013). In the presented case study, an analysis of the micrometeorology revealed that the specific topographic setting affects the energy balance at the ice field surface. Topographic shading by the steep rock face involves low solar radiation and a local minimum of air temperatures at the ice field compared to near-by weather stations, particularly in late summer. The

existence of a katabatic flow system over the ice field could be evidenced by measuring air and surface temperature fields. The temporal and spatial dynamics of the signature of the flow system revealed by the surface temperature maps suggests two different cold air flows during daytime. First, a katabatic flow draining down the ice field surface and a very shallow cold air outflow originating from the ice field caving system downstream of the ice field snout.

Numerical analysis of the flow field provided further insight into the potential origin of the katabatic flow over the ice field and its effect on heat exchange processes by driving the magnitude and the spatial distribution of sensible turbulent heat fluxes at the Ice Chapel.  During situations when a katabatic flow development  in the sun-shaded parts of the surrounding rock face was simulated, a well-developed katabatic flow drained down the ice field inducing strong turbulent mixing and turbulent sensible heat exchange towards the ice surface.

Contrary, numerical results suggest lateral atmospheric decoupling to take place in case of the onset of a shallow katabatic flow directly over the ice field suppressing heat exchange towards the ice field surface. The augmenting effect on the turbulent heat flux at the central part of the ice field in particular, might explain minimum snow ablation evidenced by measurements for this area. These results agree with earlier numerical results on boundary layer development over large snow patches (Mott et al. 2015, 2017) suggesting that the

isolation effect of shallow katabatic flows strongly affect the surface energy balance by decoupling the near-surface atmospheric layer from the adjacent warmer air. Similarly, Shea and Moore (2010), suggested that katabatic flows lower the climatic sensitivity of glaciers to external temperature changes.

This study highlights the importance of accumulation processes and micrometeorology for the survival of very small glaciers and perennial ice fields. The combination of strong snow accumulation in winter and suppression

of ice ablation in summer explain the existence of the Ice Chapel at this low elevation till today. The assessment of the relative contribution of accumulation versus ablation processes is difficult. Results suggest that the existence of the ice field is mainly a function of snow deposition by avalanches, but micrometeorological processes changing the local air temperature field are additionally required to attenuate the total mass loss of the ice field observed during the last decades. The isolation effect of shallow katabatic winds might play a crucial

role. The survival of the perennial ice field during the next decades will, however, strongly depend on the rising snowline in the future and the sum of snowfall in the Watzmann Eastface which is gravitationally redistributed towards the ice field during winter.

We believe that a next important step would be a systematic analysis of the effect of different micrometeorological processes on the local air temperature allowing a parameterization for distributed

hydrological and energy balance models, similar to what has been done for the effect of katabatic wind systems on the local air temperature over large glaciers and related surface heat fluxes (Quinn et al., 1991; Greuell and Böhm, 1998; Oerlemans and Grisogono, 2002; Petersen et al., 2013). Furthermore, the assessment of the impact of micrometeorological drivers for the sensitivity of very small glaciers to climate change will significantly contribute to an improved assessment of climate change impacts on the distribution of Alpine glaciers in future.

These findings will not be limited to Alpine glaciers, but will also be transferable to glaciers worldwide.

**Acknowledgements**

The work was funded by Swiss National Science Foundation (Project: The sensitivity of very small glaciers to micrometeorology. P300P2_164644), by the Commission for Technology and Innovation CTI (grant 2013.0288) and by the Bavarian State Ministry of the Environment and Consumer Protection (BIAS II: TKP01KPB-66747).

We thank Annette Lotz, responsible for research activities in the *Berchtesgaden National Park* for logistically supporting field experiments in the National Park.

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

## 5 Figures

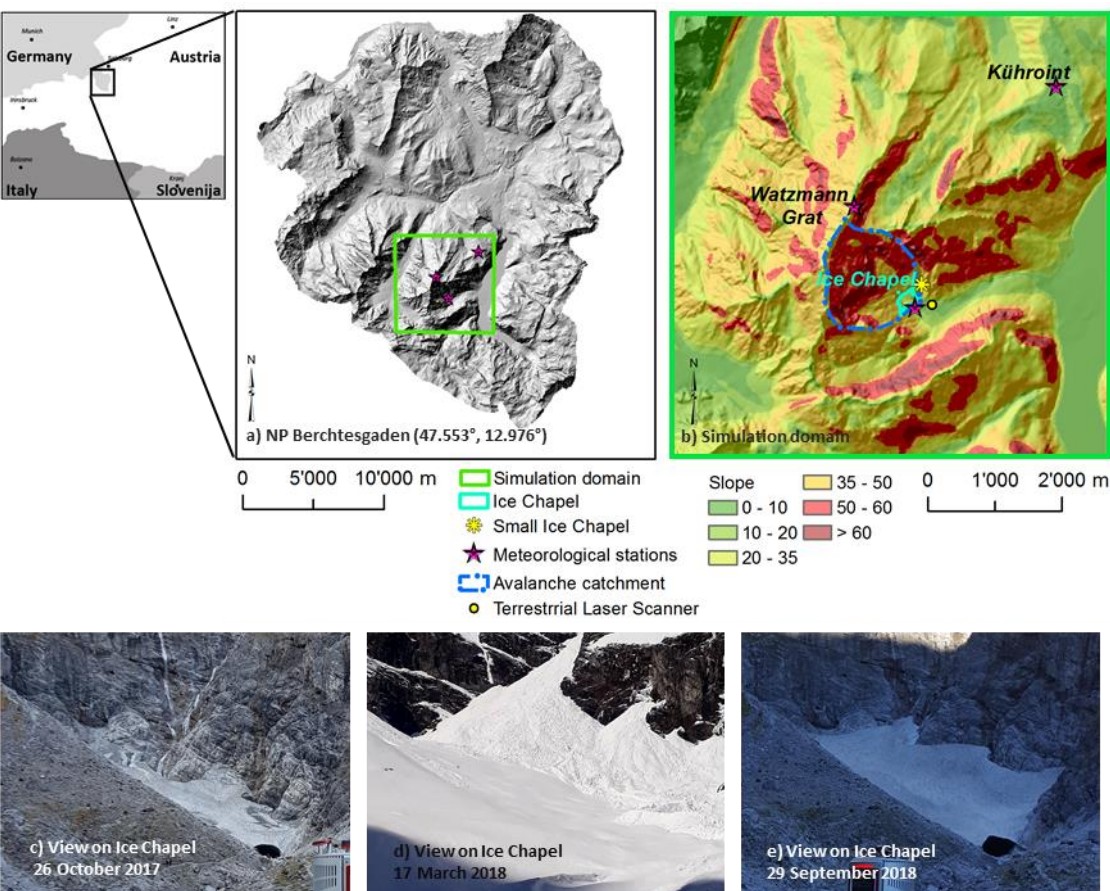

*Figure 1: a) Hillshade of Berchtesgaden National Park. b) Slope of topography shown for the simulation domain for atmospheric modelling. Meteorological stations (Stations met), the location of the large Ice Chapel and small Ice Chapel and the catchment of avalanches potentially accumulating at the large Ice Chapel is presented. c, d, e) View on the Ice Chapel from East in October 2017 from the location of Terrestrial Laser Scanner installation on 26 October 2017c) , on 17 March 2018 d) and on 29 September 2018 e).*

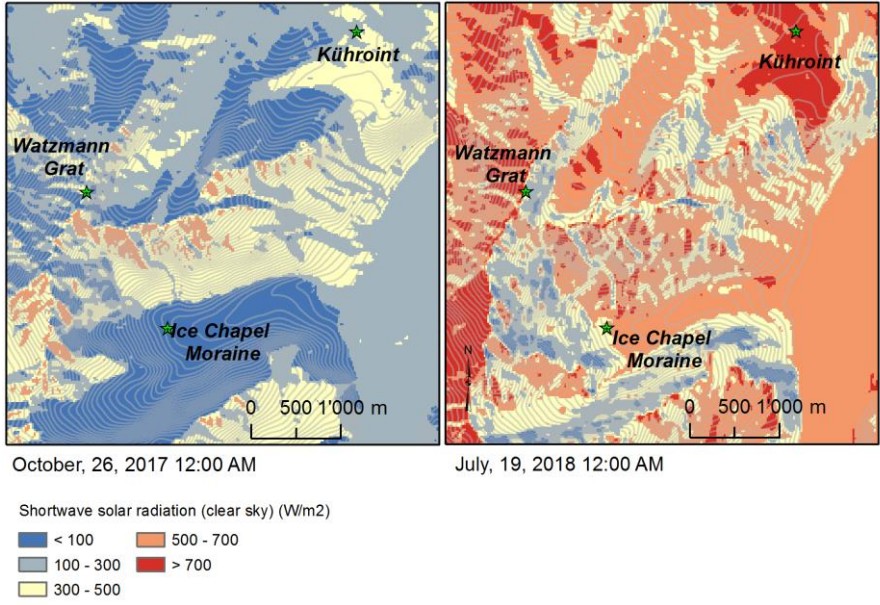

October, 26, 2017 12:00 AM

July, 19, 2018 12:00 AM

Shortwave solar radiation (clear sky) (W/m2)

- < 100
- 100 - 300
- 300 - 500
- 500 - 700
- > 700

Figure 2: Potential shortwave radiation for 26 October 2017, 12:00 PM and for 19 July 2018, 12:00 PM.

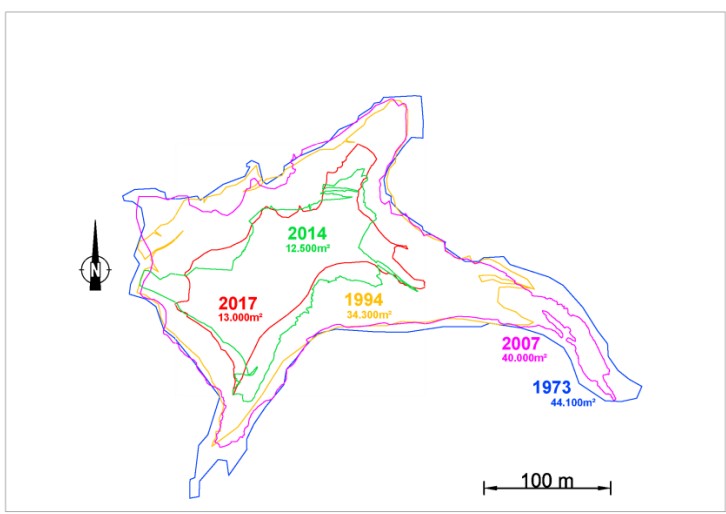

*Figure 3: Minimum surface area of the Ice Chapel measured at the end of the ablation season (October) of the respective year*

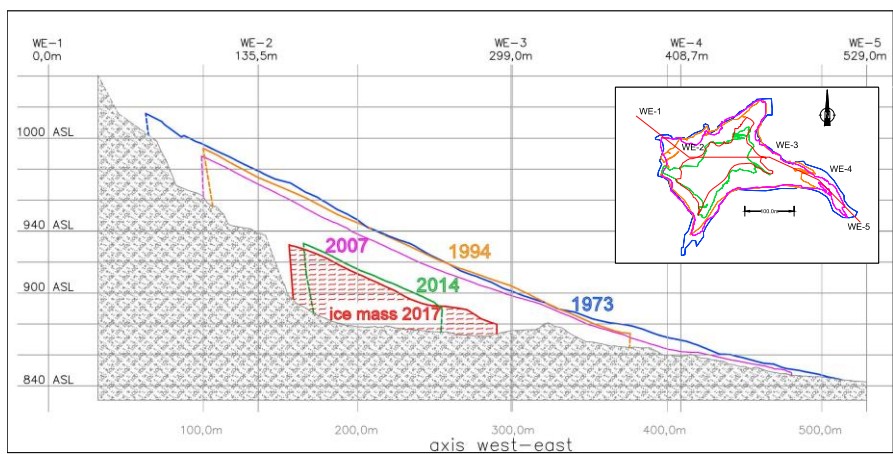

*Figure 4: Surface profile of the Ice Chapel measured at the end of the ablation season of the respective years 1973, 1994, 2007, 2014 and 2017.*

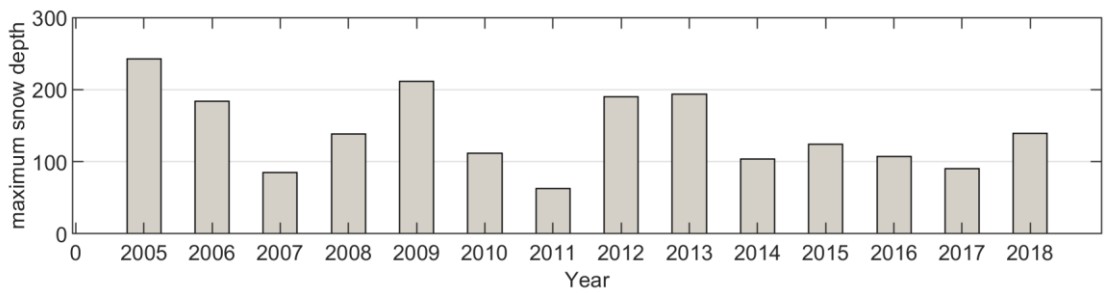

*Figure 5: Maximum seasonal snow depths measured at station Kühroint since 2005.*

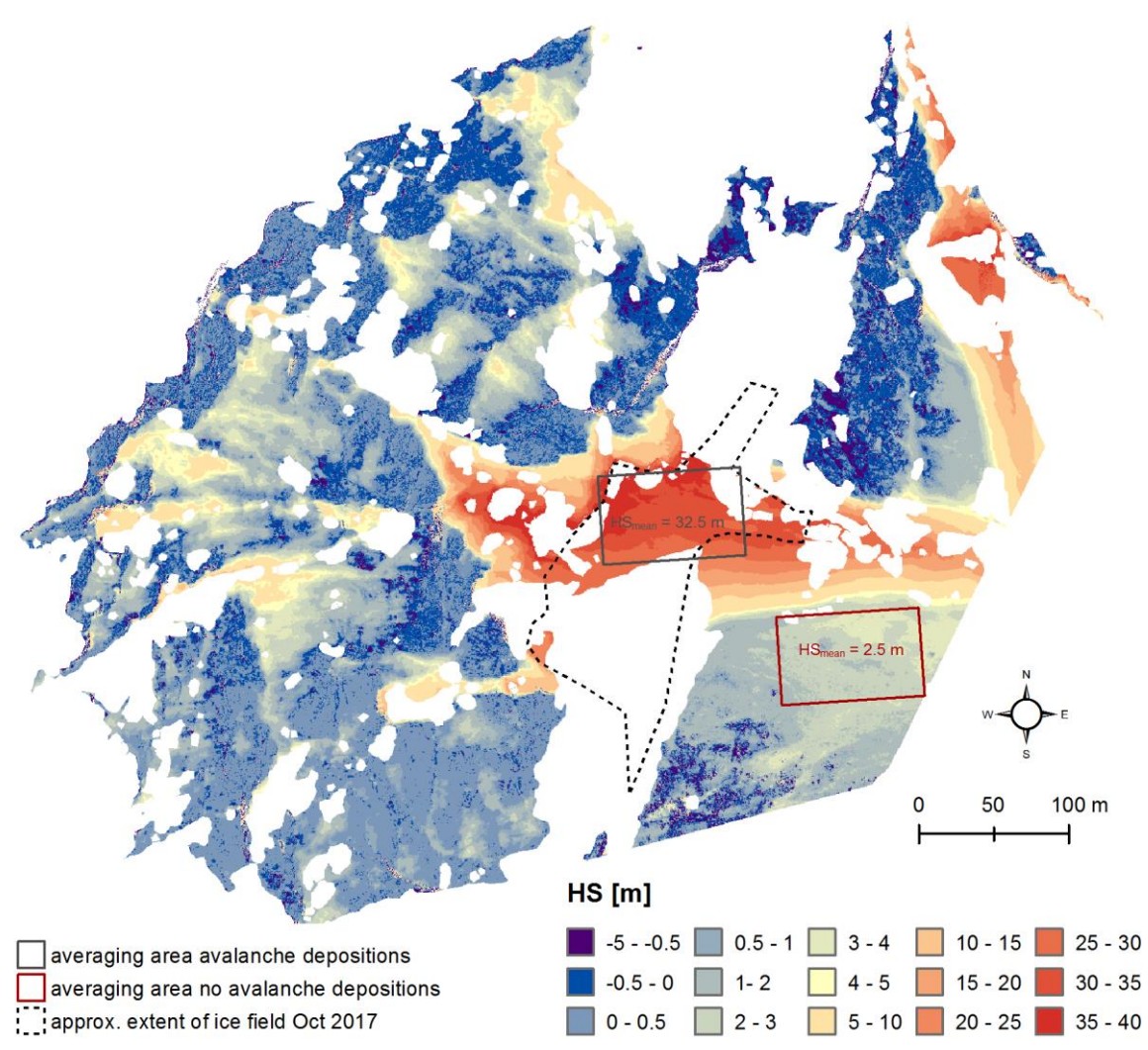

*Figure 6: Measured snow accumulation at the Ice Chapel between 26 October 2017 and 17 March 2018, obtained from Terrestrial Laser Scanning (Riegl, VZ6000). Red areas indicate high values of snow accumulation and blue colours low values. White areas represent measurement shadows. The dashed black line indicates the roughly estimated area of the Ice Chapel in October 2017. The boxes highlight averaging areas for the estimation of mean snow accumulation in areas affected by avalanching (grey box) and without avalanches (red box).*

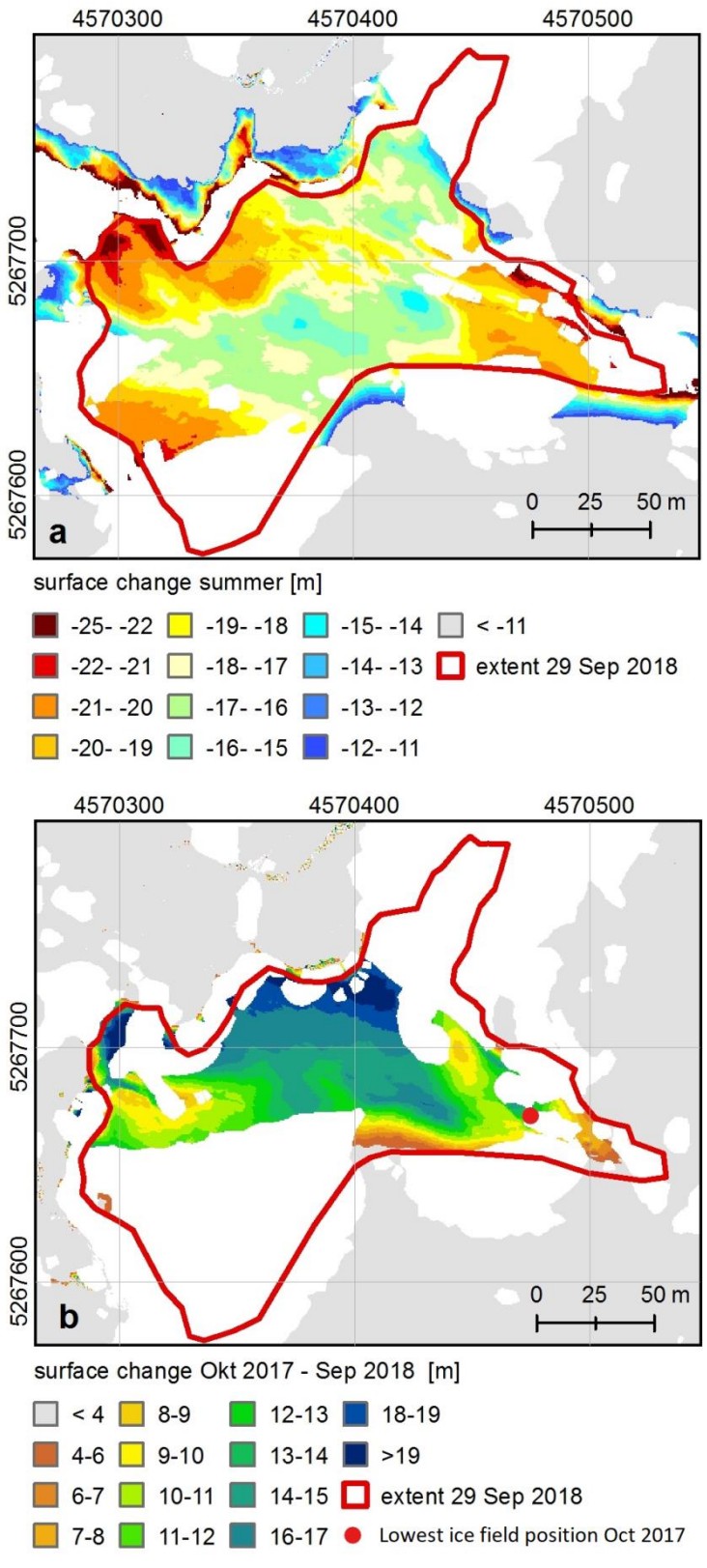

Figure 7: a) Surface change during the ablation season 2018 (March - September) b) and the surface change for the mass balance year 2017/2018 (October 2017-September 2018). The red dot marks the position of the ice field snout in October 2017.

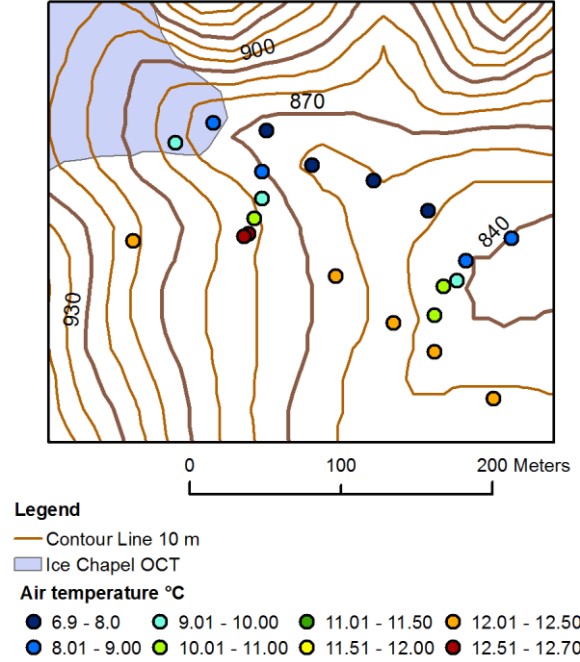

Legend
— Contour Line 10 m
▢ Ice Chapel OCT

**Air temperature °C**
● 6.9 - 8.0    ○ 9.01 - 10.00    ● 11.01 - 11.50    ○ 12.01 - 12.50
● 8.01 - 9.00    ○ 10.01 - 11.00    ○ 11.51 - 12.00    ● 12.51 - 12.70

Figure 8: Air temperatures TA measured on 26 October 2017 with mobile meteorological stations at 2 m above
5    the surface.

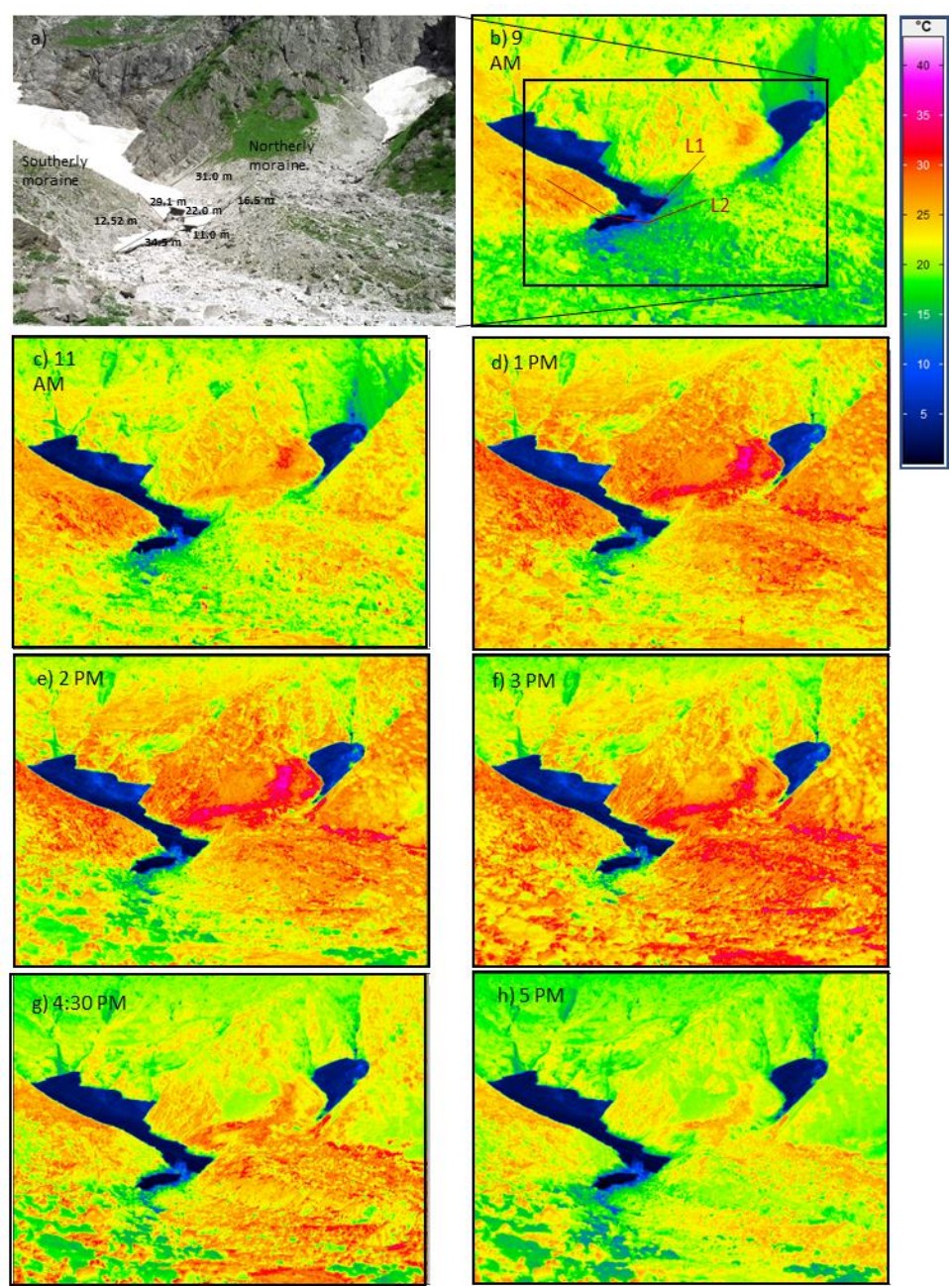

*Figure 9: Surface temperature maps obtained from IR camera on 19 July 2018 at 9 AM, 11 AM, 1 PM, 2 PM, 3 PM and 5 PM. Dark blue areas represent the ice/snow surface of the Ice Chapel and the small Ice Chapel. Values given in a) indicate measured distances in meters. Transects marked with L1 and L2 in b) show locations of transects presented in Figure 10.*

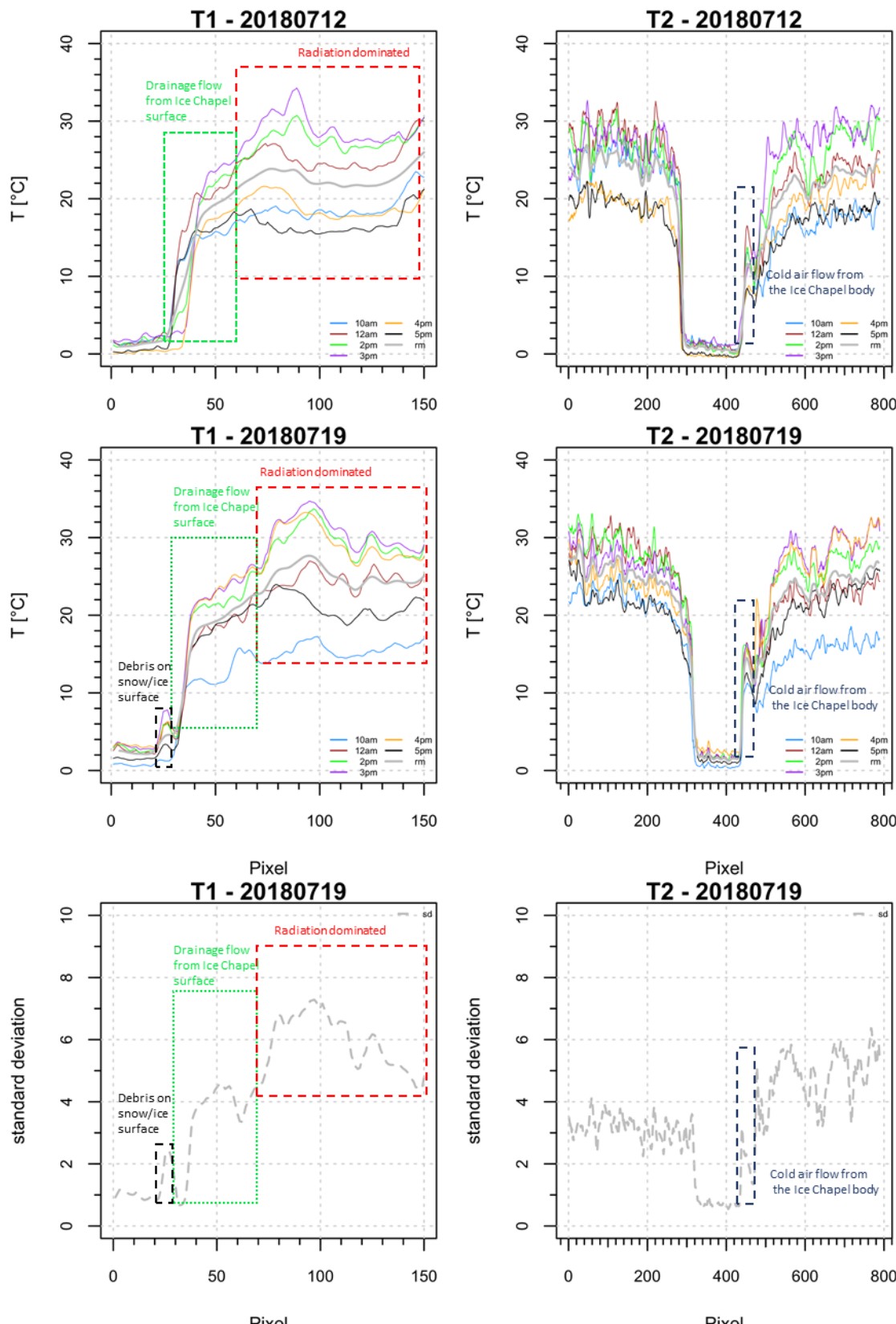

*Figure 10: Transect lines showing surface temperatures along the moraine slopes measured on 19 July 2018 at 10 AM, 12 PM, 2 PM, 4 PM and 5 PM for two measurement days and at two different locations (a-d), shown in Figure 7b. Standard deviation of TS measured at different points in time are presented in e) and f) respectively.*

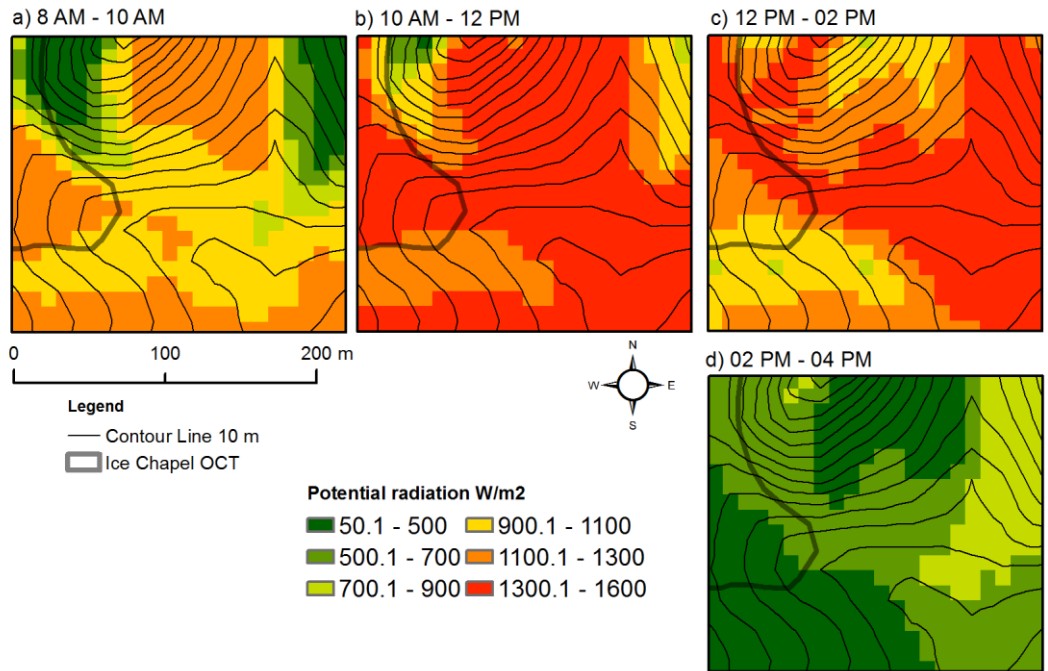

Figure 11: Potential shortwave radiation for 19 July 2018  a) 8 AM -10 AM, b) 10 AM - 12 PM, c) 12 PM - 2 PM, d) 02 PM - 4 PM.

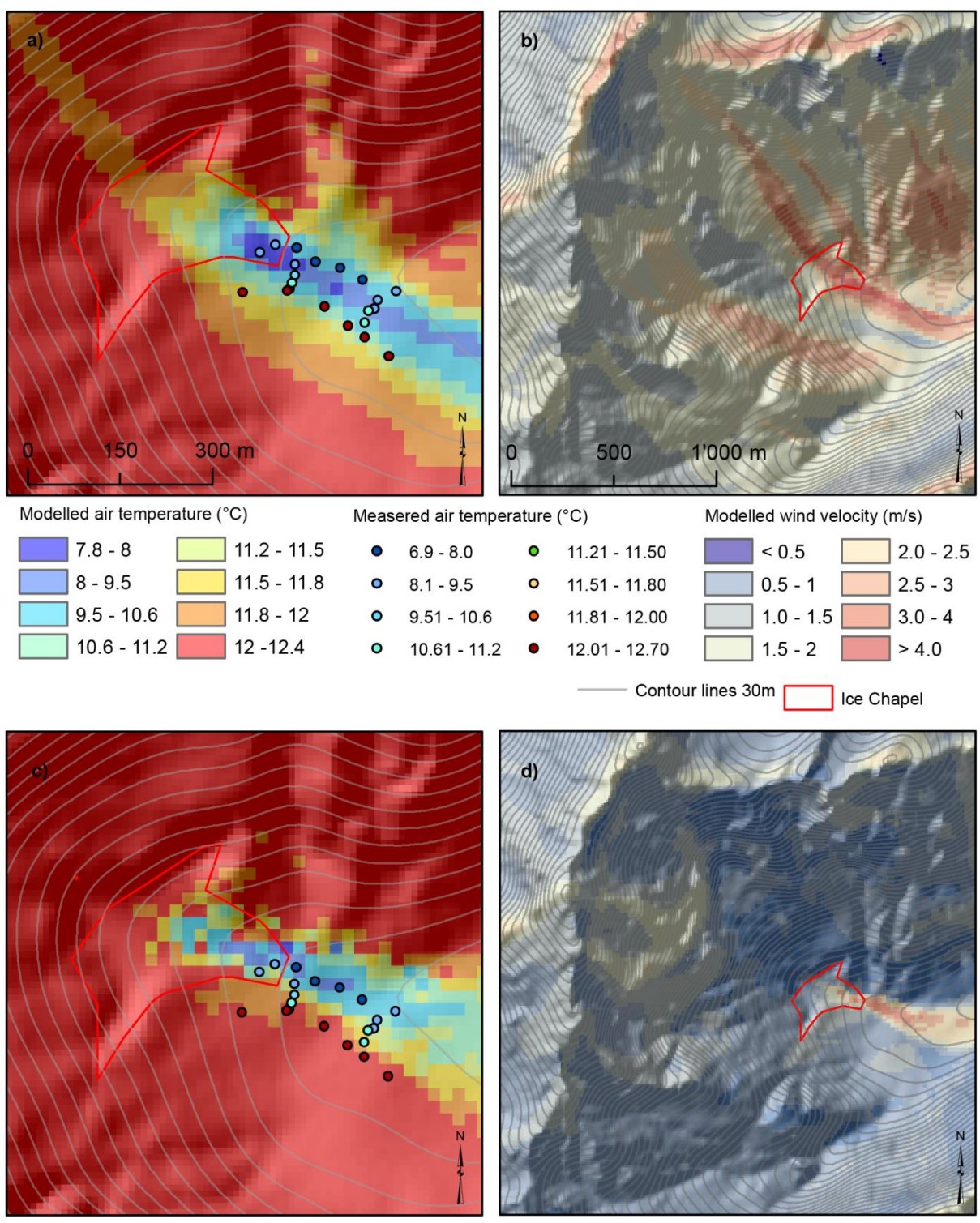

Figure 12: Air temperature and wind velocities fields modelled with the atmospheric model ARPS, initialized by a stable atmosphere (a, b) and by neutral atmosphere (c, d). Fields are shown for the first model level above ground with an average height of 2 m above the ice field surface. Simulations were initialized at 12 PM and were run for an integration time of 3600 s.

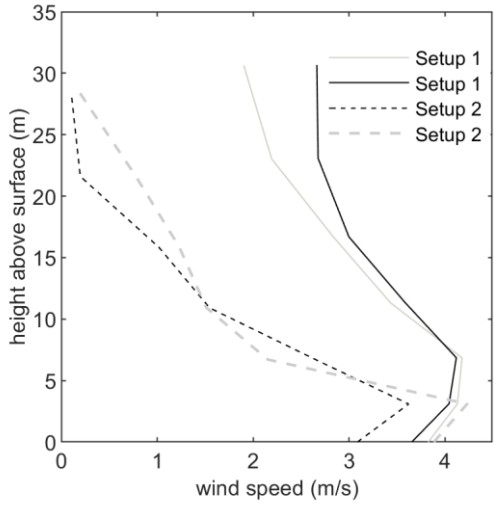

Figure 13: Near-surface profiles of wind speed for set-up 1 (solid line) and set-up 2 (dashed line) above the Ice Chapel surface. Grey lines show profiles at locations further downwind.

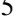
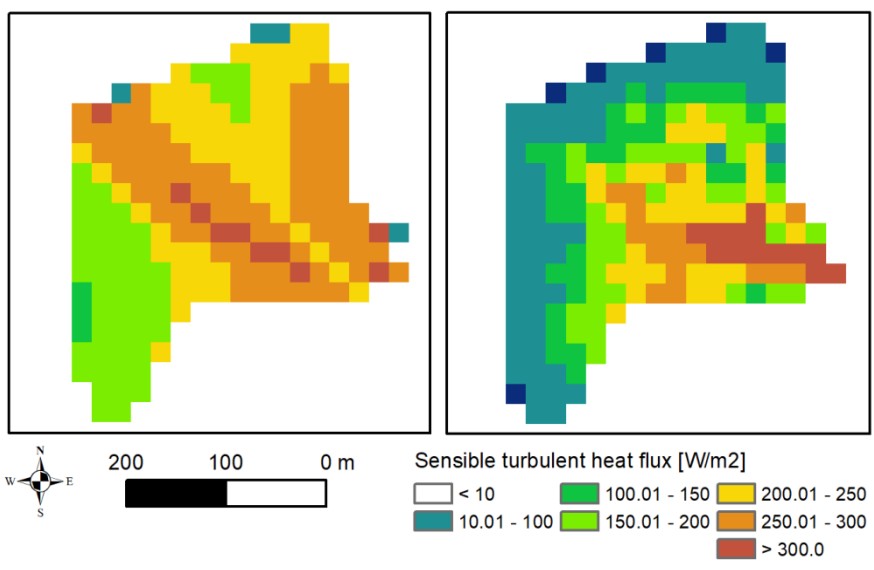

Figure 14: The sensible turbulent heat flux at the ice field surface modelled with the atmospheric model ARPS, initialized by a stable atmosphere involving a well-developed katabatic flows (setup 1, left panel) and by neutral
10    atmosphere involving a shallow katabatic flow onset at the ice field surface (setup 2, right panel).