# Peer review of "Avalanches and micrometeorology driving mass and energy balance of the lowest perennial ice field of the Alps: a case study"

_The Cryosphere, 2018_

## Referee Comment (RC1) · Kuhn (Referee) · 14 Dec 2018

The manuscript "Avalanches and micrometeorology driving mass and energy balance of the lowest perennial ice field of the Alps: a case study" by Rebecca Mott et al. gives a well written account of an exceptional set of measurements and simulations. Among others, Fig. 9 is absolutely novel and important for the understanding of the balance of such ice fields.

Other groups are presently investigating the fate of the remnants of decaying larger glaciers. In order to enable them to make use of the findings and experience of Mott and co-authors, I recommend fast publication of this paper with the following minor

changes.

Check the punctuation in the entire text. Use decimal points rather than commas. Use uniform format of the date. There is Oct 26, 2017; 29 Sep 2018; 12-09-2018... (With my restricted background I was not able to judge the performance of instruments and methods of the surveys mentioned in 2.2.2.) As I am not sure whether the deposits around the ice field would all be considered as moraines by strict geomorphologists, I suggest you state something like "we refer to moraines as any rocks that have been moved in, on top of, or under the ice field and have been deposited close to it"

In particular Page 2 / Line 8: As an example of the size distribution of glaciers in Austria quote M. Kuhn, A. Lambrecht, J. Abermann, G. Patzelt, G. Gross 2012: Austrian Glaciers 1998 and 1969: Area and Volume Changes. Zeitschrift für Gletscherkunde und Glazialgeologie 43/44 (2009/2010), 3 – 107. 2/24: 1995 Table 1: in this context, can you mention whether any of the parameters used needs to be known to better than +- 10%, +- 1 degree, or +- 0.1m? 5/12: varying by less 6/15 An atmospheric 6/28 generates 7/24 Add results given here either to Table 1 or insert a new table here, including volume change. 8/6 in winter 2017/2018 (Figure 6) was obtained 8/7 snow depth at 8/14 on this day 8/18 the Small Ice Chapel 8/19 of snow deposition. With less than 0.5 m snow accumulation, most areas 8/22 Areas which are not prone to avalanches, such as 8/31 Is some of that accumulation attributed to "solid precipitation" possibly contributed by airborne powder avalanches? 9/6 – 24 This is awkward to read. Can you summarize these changes in a small table, maybe using months instead of seasons, like "October 2017 and September 2018" in line 8, and shorten the text? 10/5 Maps of air temperatures 2 m above the surface 10/7 present. Under uniform solar radiation, measurements evidence 10/12 resulting in an air 10/22 using an IR camera 10/27 It is "Grudzielanek" in the reference list. 3.3.2 Can you estimate the so called free air temperature at the ice field elevation from the records at Kühroint (1420 m)? 13/34 Mention strong IR Radiation from the surrounding rocks. 14/14 change 14/20 ice fields is due to the 14/27 involves 16/37 year of appearance? 17/51 Zeitschrift für

Gletscherkunde und Fig.1 A larger map, e.g. including Salzburg, would be helpful. Fig. 1 Shift the legend of the middle panel on the right to the white background Fig. 3 The areas under each year are barely legible. Fig. 8 The color code is difficult to distinguish. Fig. 9 Explain the numbers in the upper left panel. Fig. 9 The color code on the right margins is difficult to read. Can you present just one at a larger scale? Fig. 10 Why is the dashed line limiting the cold air flow in the panels 10 d and 10 f not at the same x-direction? Fig.11 Maybe 50.5 – 500 and 500.1 – 700 would be sufficiently accurate. Fig.11 Which of the panels belongs to which date?

Best wishes, Michael Kuhn
* * *

---

## Referee Comment (RC2) · Fitzpatrick (Referee) · 18 Jan 2019

This paper presents an interesting study on the accumulation and ablation processes of a small mountain ice field, and the factors which may be contributing to its current survival at relatively low elevations. The authors highlight the importance of snow avalanching to the winter mass balance of the ice field, driven by the steep topography surrounding the ice field which captures and funnels snow to its surface. The authors also present a novel use of thermal imagery and boundary layer flow modelling to infer the micrometeorological conditions over the ice field, and to investigate the influence of katabatic air flow on surface ablation.

[Figure]

This work will provide a useful case study to those interested in the surface processes of very small glaciers and ice fields, and in the current and future mass balance response of ice features existing below the climatic snow line. Therefore, I believe this paper should be published in The Cryosphere. Prior to publication, I recommend a number of minor to moderate edits and revisions, which are detailed below.

My primary recommendation is in relation to the boundary layer and katabatic flow findings presented in this paper. While the thermal imagery and flow model provide an interesting overview of the potential boundary layer conditions and turbulent heat flux, I believe the paper currently lacks sufficient detail or findings to support some of the specific claims made regarding the katabatic depth, flow decoupling, and turbulent heat exchange. This could potentially be improved by providing more detail on the methods and further model output which supports these findings. If the authors feel that such detail is not available, then I recommend scaling back on these claims (some specific examples are given in the detailed comments). On a more minor note, although the paper is well written and structured, there are some punctuation issues, and the phrasing at times can hamper the understanding of the point being made, and in some cases, is incorrect. I have outlined some of these, but recommend the authors fully review the text.

Detailed Comments:

P1L26-27: Meaning of this sentence is a little hard to decipher, and it feels like two points have been mixed into one; perhaps rephrase or split into two.

P1L34: 'of the onset' and the comma could be removed.

P2L9: 'them to significant'; remove 'to'.

P2L14: 'the anomalous'; remove 'the'.

P2L23: change 'have' to 'has' or 'effect' to 'effects'.

P2L31: perhaps move 'also' to before 'expected'.

P3L1: 'is assumed to be reasoned by'; rephrase, maybe to 'is assumed to be attributable to'.

P3L18: Perhaps change 'in the centre of' to ' in the interior of', which would appear to agree more with its location on the map in Figure 1.

P3L26: 'in the angle of the rock face, the Ice Chapel'; rephrase.

Section 2.2.1: The authors have provided references to previous studies where mass balance measurements from TLS have been calculated, along with one or two details on their own measurements. I think this section would benefit from a brief but more structured description of the step-by-step process undertaken in this study to obtain the TLS measurements, the calculations to go from TLS measurements to the surface raster to the mass balance estimates, and the associated uncertainties in their values.

P3L36: Insert comma after '(Figure 2)'.

P4L3: Change 'season' to 'seasons'.

P4L6: Add units for '0.002 and 0.05'.

P4L14: Change 'extend' to 'extent'.

P4L15: Change 'allowed to' to 'has allowed for a'.

P4L17: Change 'to retrieve' to 'for the retrieval of'.

P4L17: 'surrounding' to 'surroundings', and comma after Ice Chapel.

P4L23: Change '0,01' to '0.01'.

P4L24: Add bracket before 'LEICA'.

P4L25-26: 'was checked and updated to changes applying the bundle block procedure'. I was uncertain of the meaning of this; consider rephrasing or expanding.

P5L6: At what height above the surface were the measurements taken? Was this

consistent at each location?

P5L7: '12:00 AM' should be 12:00 PM (noon). This is repeated at a number of locations in the text and in Figures 2 and 10.

P6L15: 'An' instead of 'A'.

P6L18: What is the near-surface vertical resolution of the model? Over steep terrain, the wind maximum of a katabtic flow can occur at very low heights, potentially below 2 m, as has been noted in previous studies (e.g. Denby, 1999; Oerlemans and Grisogono, 2002). Are the authors satisfied that the vertical resolution and lowest level of the model are sufficient for representing katabatic flow in these conditions? A line or two on this would be useful.

P6L28: 'generates' instead of 'generate', and 'are likely' instead of 'is likely'.

P6L35: units for '1420'.

P7L1-4: Clarification on what is your measure of atmospheric stability is required. I would be wary of describing the difference in two air temperature measurements, approximately 2 km apart, as an indicator of atmospheric stability, particularly localized stability. What form of temperature sensor and radiation screen is used at the Kühroint station? Could radiative heating of the sensor at the 'sun-exposed' site be contributing to the higher temperature? Radiative biases in temperature sensors can exceed 2°C, particularly close to noon.

P7L5: Can you define 'slightly stable' in terms of the model set up?

P7L25: Change to 'in the autumn'.

P7L34: 'Strongest reduction of the surface is…'; perhaps add in 'area' after 'surface' to clarify.

P8L35: Add 'of' after 'downstream', and change 'is' to 'are' after leeward slopes.

P9L8: Change 'made' to 'provided'.

P9L13: Remove 'rather' or quantify.

P9L23: Change 'is revealed' to 'was observed'.

P10L5: Perhaps change '12:00' to '12:00-13:00'.

P10L7: Change 'at' to 'on'.

P10L12: Change to 'resulting in an'.

P10L22: Change to 'using an IR camera'.

P10L33: Figure 9 does not have letter labels.

P10L38: Change to 'surroundings', and add comma after 'In the morning hours'.

P10L39: Add comma after 'After 2 PM'.

P12L21-23: How deep are the katabatic flows? Can you provide detail from the model that indicates the depth of the flow and its level of development?

P12L31: Are you referring to the height of the katabatic wind maximum? What height range does the model give for this? How do you know it is overestimated without comparable observations?

P12L40: Again, what is the depth of the katabatic flow?

P13L16: 'well-developed stable layers'; perhaps rephrase this to well-developed katabatic layers, as the development of a stable layer in many environments would not lead to stronger turbulent heat flux.

P13L19-21: 'Turbulent heat fluxes were shown...'; While there may be theory and evidence from previous studies to support this, I do not believe the necessary observation or model data have been presented in this paper to state if and when decoupling of the near-surface layer has occurred (and associated effects). I suggest providing more

evidence of this, if available, from the model (although this will be difficult if vertical resolution is low), or stepping back from saying this has been shown.

P13L27: Change to 'part of the ice field', and 'ablation rates with downwind distance'.

P13L28-30: How big of a role might localised shading effects play in these ablation patterns?

P13L32-33: Have you considered the effect of increased longwave radiation in areas adjacent to the warmed rock face/slopes?

P14L9: Typo 'change'.

P14L20-22: Phrasing makes the meaning a little unclear; perhaps rephrase to '…the existence of the perennial ice fields is due to the anomalous…'.

P14L32: How do you define 'very shallow' and has this really been shown?

Figure 1: Icons for Small Ice Chapel and Meteorological stations are hard to see, and I could not see the black perimeter outlining the Ice Chapel. Consider indicating the direction of view for c-e). Also, e) is not referenced in the figure caption.

Figure 2: In the scales, change the apostrophes to commas e.g. 1,000. This is repeated in Figures 11 and 12. Also, AM should be PM.

Figure 7: I would move the position of 'a)' and 'b)' in the caption to the beginning of each description. What does the blue star represent in b)?

Figure 8: It would be useful if one or more of the contour lines had the elevation indicated on them. Figure 9: Consider adding a)-h) labels.

Figure 12: Add time of day to the caption.

Figure 13: Label a) and b). Why do you think sensible heat flux in the lower, eastern portion of the Ice Chapel appears to be more intense during neutral conditions?

References: Denby, B. (1999). Second-Order Modelling of Turbulence in Katabatic

Flows. Boundary-Layer Meteorol. 92, 67–100. doi: 10.1023/A:1001796906927

Oerlemans, J., and Grisogono, B. (2002). Glacier winds and parameterisation of the related surface heat fluxes. Tellus 54A, 440–452. doi: 10.1034/j.1600-0870.2002.201398
* * *
**TCD**

---

## Author Comment (AC1) · 28 Feb 2019

Response letter to Referee M. Kuhn

We thank Michael Kuhn very much for his comments and highly valuable suggestions. We tried to address all comments. Please find a point by point responds below.

MK: Check the punctuation in the entire text. Responds: checked.

MK: Use decimal points rather than commas. Use uniform format of the date. There is Oct 26, 2017; 29 Sep 2018; 12-09-2018... Responds: we now only use decimal points and use uniform formats for date.

MK: As I am not sure whether the deposits around the ice field would all be considered as moraines by strict geomorphologists,I suggest you state something like "we refer to moraines as any rocks that have been moved in, on top of, or under the ice field and have been deposited close to it" Responds: We added a description in section "study area".

MK: In particular Page 2 / Line 8: As an example of the size distribution of glaciers in Austria quote M. Kuhn, A. Lambrecht, J. Abermann, G. Patzelt, G. Gross 2012: Austrian Glaciers 1998 and 1969: Area and Volume Changes. Zeitschrift für Gletscherkundeund Glazialgeologie 43/44 (2009/2010), 3 – 107. Responds: We added the suggested reference

5/12: varying by less Responds: corrected

6/15 An atmospheric Responds: corrected

6/28 generates 7/24 Add results given here either to Table 1 or insert a new table here, including volume change. Responds: we added values of aerial change to table 1. We do not add any values for volume change because the uncertainty of estimates increase for early years.

8/7 snow depth at Respond: corrected

8/14 on this day Responds: corrected

8/18 the Small Ice Chapel Responds: corrected

8/19 of snow deposition. With less than 0.5 m snow accumulation, most areas Responds: corrected this sentence as suggested

8/22 Areas which are not prone to avalanches, such as Responds: reformulated this sentence as suggested

8/31 Is some of that accumulation attributed to "solid precipitation" possibly contributed by airborne powder avalanches? Responds: thank you for this thought, you are right. I

added a sentence "However, part of snow deposition in the surrounding area of the ice field could also be influenced by depositions from powder snow avalanches."

9/6 – 24 This is awkward to read. Can you summarize these changes in a small table, maybe using months instead of seasons, like "October 2017 and September 2018" in line 8, and shorten the text? Responds: we included a new table (Table 3) summarizing values of snow accumulation, ablation and net surface change. Furthermore, the text is a bit shortened and limited to main results.

10/5 Maps of air temperatures 2 m above the surface Responds: changed the sentence accordingly

10/7 present. Under uniform solar radiation, measurements evidence Responds: adapted the suggested sentence.

10/12 resulting in an air Responds: changed

10/22 using an IR camera Responds: changed

10/27 It is "Grudzielanek" in the reference list. Responds: changed

3.3.2 Can you estimate the so called free air temperature at the ice field elevation from the records at Kühroint (1420 m)? Responds: Air temperature at station Kühroint ranged between 17 and 18°C during the time period 10 AM – 4 PM. This is now mentioned in the text.

13/34 Mention strong IR Radiation from the surrounding rocks. Responds: We now also mention the terrain radiation. Other effects could be strong longwave radiation from the surrounding rock face and stronger subsidence of the surface at the upper boundaries of the ice field where the lateral crevasses are most pronounced. 14/20 ice fields is due to the Responds: changed

14/27 involves Responds: changed

16/37 year of appearance? Responds: added 1997.
**TCD**

Fig.1 A larger map, e.g. including Salzburg, would be helpful. Fig.1 Shift the legend of the middle panel on the right to the white background. Responds: We revised Figure 1 adding large-scale map showing the location of the NP and changing symbols for better readability.

Fig. 3 The areas under each year are barely legible. Responds: Areas are now also provided in Table 1.

Fig. 8 The color code is difficult to distinguish. Responds: color code is updated

Fig. 9 Explain the numbers in the upper left panel. Responds: The figure caption is now updated reading: Figure 9: Surface temperature maps obtained from IR camera on July 19, 2018 at 9 AM, 11 AM, 1 PM, 2 PM, 3 PM and 5 PM. Dark blue areas represent the ice/snow surface of the Ice Chapel and the small Ice Chapel. Values given in a) indicate measured distances in meters. Transects marked with L1 and L2 in b) show locations of transects presented in Figure 10. Fig. 9 The color code on the right margins is difficult to read. Can you present just one at a larger scale? Responds: Figure 9 is revised, now having only one but larger scale.

Fig.10 Why is the dashed line limiting the cold air flow in the panels 10 d and 10 f not at the same x-direction? Responds: Figure 10 is revised – the dashed line limiting the cold air outflow is now at the same-x-drection.

Fig.11 Maybe 50.5 – 500 and 500.1 – 700 would be sufficiently accurate. Responds: Legend of Figure 11 has been updated.

Fig.11 Which of the panels belongs to which date? Responds: there was an error in the figure caption. The figure caption now reads: Figure 11: Potential shortwave radiation for July 19, 2018 a) 8 AM -10 AM, b) 10 AM-12 PM, c) 12 PM -02:00 PM, d) 02 PM – 4 PM.

Response letter to Referee N. Fitzpatrick

We thank N. Fitzpatrick very much for his comments and highly valuable suggestions. We tried to address all comments. Please find a point by point responds below.

Major Comments: 1. My primary recommendation is in relation to the boundary layer and katabatic flow findings presented in this paper. While the thermal imagery and flow model provide an interesting overview of the potential boundary layer conditions and turbulent heat flux, I believe the paper currently lacks sufficient detail or findings to support some of the specific claims made regarding the katabatic depth, flow decoupling, and turbulent heat exchange. This could potentially be improved by providing more detail on the methods and further model output which supports these findings. If the authors feel that such detail is not available, then I recommend scaling back on these claims (some specific examples are given in the detailed comments). Responds: we agree with the referee that the combination of measurements and model results only suggest but do not evidence processes that are discussed. Additional model runs will not significantly change the main conclusion from the numerical part of the manuscript. We changed some parts of the manuscript to scale back the claims and highlight that the discussion is mainly based on model and measurement results suggesting specific feedbacks between the boundary layer flow and the energy balance. We also added a new figure showing the depth of the katabatic flow at the Ice Chapel depending on model setup.

2. On a more minor note, although the paper is well written and structured, there are some punctuation issues, and the phrasing at times can hamper the understanding of the point being made, and in some cases, is incorrect. I have outlined some of these, but recommend the authors fully review the text. Responds: we went through the text and revised multiple parts of the manuscript.

Detailed Comments:

P1L26-27: Meaning of this sentence is a little hard to decipher, and it feels like two points have been mixed into one; perhaps rephrase or split into two. Responds:

changed accordingly;

P1L34: 'of the onset' and the comma could be removed. Responds: changed the sentence.

P2L9: 'them to significant'; remove 'to'. Responds: changed accordingly;

P2L14: 'the anomalous'; remove 'the'. Responds: changed accordingly;

P2L23: change 'have' to 'has' or 'effect' to 'effects'. Responds: changed accordingly;

P2L31: perhaps move 'also' to before 'expected'. Responds: changed accordingly;

P3L1: 'is assumed to be reasoned by'; rephrase, maybe to 'is assumed to be attributable to'. Responds: changed.

P3L18: Perhaps change 'in the centre of' to ' in the interior of', which would appear to agree more with its location on the map in Figure 1. Responds: changed text as suggested.

P3L26: 'in the angle of the rock face, the Ice Chapel'; rephrase. Responds: we changed this sentence according to the reviewer suggestion.

Section 2.2.1: The authors have provided references to previous studies where mass balance measurements from TLS have been calculated, along with one or two details on their own measurements. I think this section would benefit from a brief but more structured description of the step-by-step process undertaken in this study to obtain the TLS measurements, the calculations to go from TLS measurements to the surface raster to the mass balance estimates, and the associated uncertainties in their values. Responds: We have revised the methods section now providing a more detailed description of the post processing of TLS data: In past studies, repeated TLS was successfully applied to calculate snow volumes (Grünewald et al., 2018) or snow depth changes during the accumulation (Mott et al., 2010; Schirmer et al., 2011; Sommer et al., 2015) and ablation season (Grünewald et al., 2010; Egli et al., 2011; Mott et

al., 2011; Schlögl et al., 2018) with a vertical accuracy of less than 10 cm for 300 m distance (e.g. Prokop et al., 2008; Revuelto et al., 2014). A more general description of the TLS measurement setup and accuracy over snow can be found in Prokop et al. (2008), Schaffhauser et al. (2008), and Grünewald et al. (2010). To reduce scan shadows the laser scanner was set up at up to three different positions. The area of the Ice Chapel and its surrounding was then recorded with a frequency of 300 kHz and angular step widths between 0.002 and 0.05 depending on maximum measurement distance which ranged from 300 to 500 m. We followed the post processing procedure described by Grünewald et al. (2018 and 2019): First coarse registration was performed using small reflector plates mounted in the area and/or topographic features (such as well-defined rocks) as tiepoints. This registration was then improved by applying a 3D-surface matching function (Multi station adjustment; Riegl, 2011). In the following, the data were transformed to a global coordinate system (UTM). Finally, data amounts were reduced by aggregation of the point clouds to 25 cm 3D grids (octree filter) and raster of surface change (cell size 0.5 m) were calculated in ArcMap 10.2.

P3L36: Insert comma after '(Figure 2)'. Responds: inserted a comma.

P4L3: Change 'season' to 'seasons'. Responds: changed.

P4L6: Add units for '0.002 and 0.05'. Responds: we added degree.

P4L14: Change 'extend' to 'extent'. Responds: changed.

P4L15: Change 'allowed to' to 'has allowed for a'. Responds: changed.

P4L17: Change 'to retrieve' to 'for the retrieval of'. Responds: changed.

P4L17: 'surrounding' to 'surroundings', and comma after Ice Chapel. Responds: changed

P4L23: Change '0,01' to '0.01'. Responds: changed

P4L24: Add bracket before 'LEICA'. Responds: added a bracket.
P4L25-26: 'was checked and updated to changes applying the bundle block procedure'. I was uncertain of the meaning of this; consider rephrasing or expanding. Responds: We skipped this sentence as this is not a very important information for the reader but would need more detailed information if included.

P5L6: At what height above the surface were the measurements taken? Was this consistent at each location? Responds: at 2 m above the surface;

P5L7: '12:00 AM' should be 12:00 PM (noon). This is repeated at a number of locations in the text and in Figures 2 and 10. Responds: changed the date format throughout the text and all figures.

P6L15: 'An' instead of 'A'. Responds: changed

P6L18: What is the near-surface vertical resolution of the model? Over steep terrain, the wind maximum of a katabatic flow can occur at very low heights, potentially below 2 m, as has been noted in previous studies (e.g. Denby, 1999; Oerlemans and Grisogono, 2002). Are the authors satisfied that the vertical resolution and lowest level of the model are sufficient for representing katabatic flow in these conditions? A line or two on this would be useful. Responds: we now more clearly state the range of the vertical resolution of the first grid cell above the ground and the average resolution of the first grid level at the Ice Chapel area: The vertical resolution of the first grid above surface ranges between 1.4 m and 2.6 m with an average value of approximately 2 m at the ice Chapel area. In the discussion part we refer to the limitation of model resolution for calculating shallow katabatic flows: The representation of the katabatic flow depth is known to be strongly dependent on the near-surface vertical grid resolution (Mott et al., 2015). A vertical resolution of approximately 2 m close to the surface appear to be too coarse to capture shallow katabatic flows which typically have a jet maximum of less than 2 m above ground (Denby, 1999; Oerlemans and Grisogono, 2002).

The main aim of this study is, however, not to capture the exact depth of the katabatic flow, but to investigate the different behavior in heat exchange processes with the different onset and evolution of the katabatic flow. Running the model on a horizontal resolution of 20 m is already a great challenge for steep terrain. Using a higher model resolution would produce numerical instabilities and would not allow the model for the full integration time. The evolution of the depth of the katabatic flow is now shown in Figure 13.

P6L28: 'generates' instead of 'generate', and 'are likely' instead of 'is likely'. Responds: Thank you for your very detailed corrections! We changed the two words accordingly.

P6L35: units for '1420'. Responds: added 'm'.

P7L1-4: Clarification on what is your measure of atmospheric stability is required. I would be wary of describing the difference in two air temperature measurements, approximately 2 km apart, as an indicator of atmospheric stability, particularly localized stability. What form of temperature sensor and radiation screen is used at the Kühroint station? Could radiative heating of the sensor at the 'sun-exposed' site be contributing to the higher temperature? Radiative biases in temperature sensors can exceed 2ậŮęC, particularly close to noon. Responds: we used 3 stations to obtain the initial mean values for atmospheric stability, which was Brunt‐Väisälä frequency Nstat≈0.01 s−1 in case of slightly stable conditions. We however want to highlight that the local terrain shading effects in the Watzmann East face, considered in the model, as well as the evolution of surface temperatures much stronger affects the local boundary layer evolution in the model than the initial atmospheric stability of the model. The sensor at Kühroint is a shielded temperature sensor and radiative effects are expected to be not very high because of the low radiation at this time of the season (end of October, calculated radiation is also presented in Figure 2). We added the value of atmospheric stability to the methods section and discussed the shortcoming of this approach:

We used this methodology since no direct measurements are available at the Watzmann Eastface and no meteorological measurements are available at the Ice Chapel for 19 July. Furthermore, heating of the sensors by shortwave radiation might also

affect air temperature measurements. Initial atmospheric stability is thus only an approximation of local atmospheric conditions. Since simulations are not run for 24 hour integration time the integration time does not allow for the full adaptation of the near-surface air field to the daily cycle of radiation. As discussed above, however, we expect the flow field to adapt to thermal forcing during the integration time, also changing the local atmospheric stability.

P7L5: Can you define 'slightly stable' in terms of the model set up? Responds: we initialized the atmospheric profile at base state corresponding to a Brunt‐Väisälä frequency Nstat≈0.01 s−1.

P7L25: Change to 'in the autumn'. Responds: changed

P7L34: 'Strongest reduction of the surface is...'; perhaps add in 'area' after 'surface' to clarify. Responds: added;

P8L35: Add 'of' after 'downstream', and change 'is' to 'are' after leeward slopes. Responds: changed;

P9L8: Change 'made' to 'provided'. Responds: changed

P9L13: Remove 'rather' or quantify. Responds: removed

P9L23: Change 'is revealed' to 'was observed'. Responds: changed

P10L5: Perhaps change '12:00' to '12:00-13:00'. Responds: changed

P10L7: Change 'at' to 'on'. Responds: the sentence is changed (see comment to referee 1)

P10L12: Change to 'resulting in an'. Responds: changed

P10L22: Change to 'using an IR camera'. Responds: changed

P10L33: Figure 9 does not have letter labels. Responds: we changed Figure 9, having one legend for surface temperature allowing a better readability and adding letter labels

to all figures.

P10L38: Change to 'surroundings', and add comma after 'In the morning hours'. Responds: changed

P10L39: Add comma after 'After 2 PM'. Responds: added

P12L21-23: How deep are the katabatic flows? Can you provide detail from the model that indicates the depth of the flow and its level of development? Responds: the depth of katabatic flow varies between 3 – 4 m at the Ice Chapel for setup 2 and between 7 to 9 m for setup 1. We now include a figure showing the near-surface profile of wind speed for the two setups and at two downwind distances. We are now referring to the new Figure 13 accordingly.

P12L31: Are you referring to the height of the katabatic wind maximum? What height range does the model give for this? How do you know it is overestimated without comparable observations? Responds: The depth of katabatic flow varies between 3 – 4 m at the Ice Chapel for setup 2 and between 7 to 9 m for setup 1. We compare model results with measured proxy of air temperature via IR measurements. While measurements show a much smaller horizontal extend of the area cooled by the presence of the drainage flow, model results show a larger area. The reviewer is totally right that the statement of an overestimated height of the katabatic flow is only deduced from this comparison but not directly evidenced by measurements. We changed the text to be more clear about this.

P12L40: Again, what is the depth of the katabatic flow? Responds: the depth of katabatic flow varies between 3 – 4 m at the Ice Chapel for setup 2 and between 7 to 9 m for setup 1.

P13L16: 'well-developed stable layers'; perhaps rephrase this to well-developed katabatic layers, as the development of a stable layer in many environments would not lead to stronger turbulent heat flux. Responds: yes, the referee is totally right – we changed
the sentence accordingly.

P13L19-21: 'Turbulent heat fluxes were shown...'; While there may be theory and evidence from previous studies to support this, I do not believe the necessary observation or model data have been presented in this paper to state if and when decoupling of the near-surface layer has occurred (and associated effects). I suggest providing more evidence of this, if available, from the model (although this will be difficult if vertical resolution is low), or stepping back from saying this has been shown. Responds: we changed this paragraph now reading: Modelled turbulent heat fluxes (Figure 14) were smaller in case of weaker and more shallow drainage flows due to a decoupling of the atmospheric layer adjacent to the ice field surface from the warmer air above. These model results are similar results as discussed in Mott et al. (2015) who emphasized the isolation effect of shallow katabatic winds over large snow fields, also referred to as lateral atmospheric decoupling. Model results suggest that in such situations, the snow and ice melt is only marginally affected by higher ambient air temperatures. We do not refer to measurements here since no turbulence measurements are available. We only discuss model results and set these in context to earlier studies.

P13L27: Change to 'part of the ice field', and 'ablation rates with downwind distance'. Responds: changed accordingly

P13L28-30: How big of a role might localised shading effects play in these ablation patterns? Responds: since these pattern of smaller ablation rates are found at the central part of ice field and given the rather constant aspect of the ice field (see also Figure 2 - modeled radiation) we do not expect that shading effect explain these local patterns.

P13L32-33: Have you considered the effect of increased longwave radiation in areas adjacent to the warmed rock face/slopes? Responds: we now also refer to possible effects due to longwave radiation (see also response to referee 1): Maximum snow ablation rates at high elevated areas, however, cannot be explained by modelled flow

field dynamics. One reason for above-average snow ablation in this region might be the larger amount of debris accumulated at the boundary areas adjacent to the rock face and the moraine slopes. Other effects could be strong longwave radiation from the surrounding rock face and stronger subsidence of the surface at the upper boundaries of the ice field where the lateral crevasses are most pronounced. P14L9: Typo 'change'. Responds: changed

P14L20-22: Phrasing makes the meaning a little unclear; perhaps rephrase to '...the existence of the perennial ice fields is due to the anomalous...'.

P14L32: How do you define 'very shallow' and has this really been shown? Responds: measurements, i.e. Figure 10 strongly indicate a shallow katabatic flow from the ice field body. Temperature measurements suggest a shallow katabatic flow – we do not say that these proxy evidence a shallow katabatic flow.

Figure 1: Icons for Small Ice Chapel and Meteorological stations are hard to see, and I could not see the black perimeter outlining the Ice Chapel. Consider indicating the direction of view for c-e). Also, e) is not referenced in the figure caption. Responds: we revised Figure 1 improving the readability of symbols. We additionally indicate the location of the TLS and reference e in the figure caption.

Figure 2: In the scales, change the apostrophes to commas e.g. 1,000. This is repeated in Figures 11 and 12. Also, AM should be PM. Responds: we revised Figures 2, 11 and 12.

Figure 7: I would move the position of 'a)' and 'b)' in the caption to the beginning of each description. What does the blue star represent in b)? Responds: we changed the figure. The red dot now marks the position of the ice field snout in October 2017. We removed the blue star.

Figure 8: It would be useful if one or more of the contour lines had the elevation indicated on them. Responds: We revised figure 8 now showing labels every 30 m.

Figure 9: Consider adding a)-h) labels. Responds: labels added

Figure 12: Add time of day to the caption. Responds: we added: Simulations were initialized at 12 PM and were run for an integration time of 3600 s. Figure 13: Label a) and b). Why do you think sensible heat flux in the lower, eastern portion of the Ice Chapel appears to be more intense during neutral conditions? Responds: Turbulent heat fluxes are higher at the lowest part of the ice chapel during neutral conditions because of a higher mean near-surface air temperature at the ice field for this situation and an increase in wind velocity towards the ice field snout at the same time. For more stable conditions, cooler air is advected from higher elevated sites towards the ice field decreasing the local air temperature there. Stronger near-surface mechanical turbulence, however, results in a more efficient heat exchange towards the ice field surface for situations with well-developed katabatic flows. To clarify this we changed the paragraph reading: Although turbulent fluxes are locally higher at the lower parts due to higher mean near-surface air temperatures than for the well-developed katabatic flow situation, the average turbulent sensible heat flux at the ice field surface is significantly smaller (Figure 14). This confirms earlier results presented by Mott et al. (2015) showing stronger turbulent heat fluxes in situations with well-developed katabatic flows compared to shallow katabatic flows. The strong katabatic winds enhance mechanical turbulence close to the surface and remove the shallow stable layer close to the ice surface that typically promote a suppression of turbulent heat exchange. Modelled mean turbulent heat fluxes at the ice field surface (Figure 14) were smaller in case of weaker and more shallow drainage flows due to a decoupling of the atmospheric layer adjacent to the ice field surface from the warmer air above. These model results are similar to results discussed in Mott et al. (2015) who emphasized the isolation effect of shallow katabatic winds over large snow fields, also referred to as lateral atmospheric decoupling. Model results suggest that in such situations, the snow and ice melt is only marginally affected by higher ambient air temperatures.

We added References: Denby, B. (1999). Second-Order Modelling of Turbulence in Katabatic Flows. Boundary-Layer Meteorol. 92, 67–100. doi: 10.1023/A:1001796906927 Oerlemans, J., and Grisogono, B. (2002). Glacier winds and parameterisation of the re- lated surface heat fluxes. Tellus 54A, 440–452. doi: 10.1034/j.1600-0870.2002.20139